**Implementation of global soil databases in NOAH-MP model and the effects on**

**simulated mean and extreme soil hydrothermal changes**

Kazeem A. Ishola[1,3] Gerald Mills[2], Ankur P. Sati[2], Benjamin Obe[2], Matthias Demuzere[5],

Deepak Upreti[1,3], Gourav Misra[3], Paul Lewis[3], Daire Walsh[3], Tim McCarthy[3], Rowan

Fealy[4,*]

[1]Irish Climate Analysis and Research UnitS (ICARUS), Maynooth University, Maynooth, Ireland

[2]School of Geography, University College Dublin, Dublin, Ireland

[3]National Centre for Geocomputation, Maynooth University, Maynooth, Ireland

[4]Department of Geography, Maynooth University, Maynooth, Ireland.

[5]B-Kode VOF, Ghent, Belgium
*Correspondence to:* Kazeem Ishola (Kazeem.Ishola@mu.ie) and Rowan Fealy (Rowan.Fealy@mu.ie)

**Abstract**

Soil properties and their associated hydro-physical parameters represent a significant

source of uncertainty in Land Surface Models (LSMs) with consequent effects on

simulated sub-surface thermal and moisture characteristics, surface energy

exchanges and turbulent fluxes. These effects can result in large model differences

particularly during extreme events. Typical of many model-based approaches, spatial

soil information such as location, extent and depth of soil textural classes are derived

from coarse scale soil information and employed largely due to their ready availability

rather than suitability. However, the use of a particular spatial soil dataset can have

important consequences for many of the processes simulated within a LSM. This study

investigates model uncertainty in the NOAH-MP model in simulating soil moisture

(expressed as a ratio of water to soil volume, $m^3$ $m^{-3}$) and soil temperature changes,

associated with two widely used global soil databases (STATSGO and SOILGRIDS).

Both soil datasets produced a significant dry bias in loam soils, of 0.15 $m^3$ $m^{-3}$ and 0.10

$m^3$ $m^{-3}$ during a wet and dry period, respectively. The spatial disparities between

STATSGO and SOILGRIDS also influenced the simulated regional soil hydrothermal

changes and extremes. SOILGRIDS was found to intensify drought characteristics -

shifting low/moderate drought areas into the extreme/exceptional classification -

relative to STATSGO. Our results demonstrate that the coarse STATSGO performs as

good as the fine-scale SOILGRIDS soil database, though the latter represents the soil

moisture dynamics better. However, the results underscore the need to develop

detailed regionally-derived soil texture characteristics and for better representations of

soil properties in LSMs to improve operational modeling and forecasting of hydrological

processes and extremes.

Keywords: soil moisture; soil temperature, droughts; Land surface model; soil hydro-

physical properties

# 1.    Introduction

The pedosphere (or soil) is an important component of the Earth system and plays a critical role in energy, water and biogeochemical exchanges that occur at the land-atmosphere interface (Dai et al., 2019a;b). The accurate description and representation of soil textural categories and/or soil hydro-physical properties is fundamental to developing and enhancing Earth system modeling (ESM) capacity in predicting land surface exchanges at different scales (Luo et al., 2016; Dai et al., 2019a,b). This information is incorporated via the respective land surface model (LSM) – the only physical boundary in an ESM - and a key component of any ESM framework (Fisher and Koven, 2020; Blyth et al., 2021). However, accurate descriptions of soil properties in LSMs are difficult to obtain due to the limited availability of high resolution global-scale soil texture measurements or lack of regionally specific measured soil properties (e.g. Kishné et  al. 2017; Dennis and Berbery, 2021; 2022). This represents a key limitation and is a source of model uncertainty in current LSMs (Li et al., 2018; Zhang et al, 2023), and consequently weather and climate models.

In many LSMs, soil hydrothermal properties such as soil thermal and hydraulic conductivity and diffusivity, porosity, field capacity, wilting point, saturated soil matric potential, etc. are linked to soil textural classes/composition in one of two ways. Typically, models employ a model-prescribed look-up table, with values that are derived from often limited (e.g. geographically and data limited) in-situ soil surveys, to associate mean or typical soil properties with each soil category. The soil categories are identified by grouping soil samples with similar properties using particle size analysis (e.g. Gee and Bauder, 2018). While this option is computationally efficient, it relies on the assumption that the derived values are transferable; this is not likely to be realistic as soil properties vary, depending on parent materials, climate, age, management etc. This approach is also dependent on having access to soil texture maps; the accuracy, scale and extent of which varies between different soil databases (Zhao et al., 2018; Dai et al., 2019a,b; Dennis and Berbery, 2022). In spite of this, the use of readily available global soil texture maps, in combination with model look-up tables, is a standard practice in ESM research.

As an alternative, new state-of-the-art global soil information datasets are being explored to constrain and potentially improve the representation of soil processes within LSMs (e.g. de Lannoy et al., 2014; Shangguan et al., 2014; Hengl et al., 2017; Looy et al., 2017; Dennis and Berbery, 2021;2022; Xu et al., 2023). For example, soil hydro-thermal properties can be estimated from a set of equations known as PedoTransfer Functions (PTFs) that require information on soil such as sand, silt and clay composition and organic matter (Looy et al., 2017; Dai et al., 2019a,b). PTFs have been derived based on a variety of different approaches (Looy et al., 2017) including,

physically-based relationships or advanced statistical approaches using machine learning, random forest and neural networks (Lehmann et al., 2018; Zhang et al., 2018; Or and Lehmann, 2019; Szabó et al. 2019) and vary in complexity. While the choice of PTFs partly depend on the availability of inputs, (Weihermüller et al., 2021) they have been reported to impact soil moisture simulations, with consequent effects on the surface energy and water fluxes, land-atmosphere coupling, atmospheric moisture budget, boundary layer evolution and simulation of regional climates (e.g. Dennis and Berbery, 2021; 2022; Weihermüller et al. 2021; Xu et al., 2023; Zhang et al., 2023).

Moreover, as soil moisture affects land-atmosphere interactions, largely through its control on the evaporative fraction (e.g. Seneviratne et al., 2010; Ishola et al., 2022), soil hydro-physical properties play an important role in determining the land surface response to climate extremes (e.g. droughts) (He et al., 2023; Zhang et al., 2023). Weihermüller et al. (2021), using the HYDRUS-1D model, reported that soil hydraulic properties estimated from different PTFs resulted in substantial variability in model estimated water fluxes. In this context, Dennis and Berbery (2021) and Dennis and Berbery (2022) employed soil properties derived STATSGO and the Global Soil Dataset for Earth System Modelling (GSDE), in both the Weather and Research Forecasting (WRF) and Community Land Model (CLM) models. They found soil texture-related differences in the surface fluxes that could lead to differences in the evolution of boundary layer thermodynamic structure and development of precipitation, findings consistent with Zhang et al. (2023). The use of new soil information, such as POLARIS and the 250 m SOILGRIDS, has been found to improve the performance of LSMs (Xu et al., 2023), but based on a limited number of studies.

Zhang et al. (2023) was one of the first to implement SOILGRIDS in the coupled WRF Hydrological Modelling system (WRF-Hydro), of which NOAH-MP is the land surface model, to evaluate the role of four different global soil datasets on land atmosphere interactions over Southern Africa. While Zhang et al. (2023) found that the ensemble of model simulations, based on the different soil data inputs, was able to reasonably reproduce the spatial, and spatio-temporal, patterns of the surface hydrometeorological fields investigated, soil texture differences, specifically those associated with differences in soil properties, were found to directly impact model estimated soil moisture, with associated impacts on skin and air temperature and sensible heat fluxes. Importantly, for the study and domain outlined here, the effects of different soil texture datasets on soil moisture were found to decrease with increasing aridity (Zheng and Yang, 2016; Zhang et al., 2023). Consequently, the authors highlighted the need to consider study location and background climate in addition to the different schemes for estimating soil hydro-thermal processes. While it is widely recognized that LSMs will respond to changes in other drivers, such as

vegetation (e.g. albedo, surface roughness length, etc.) and meteorological forcing (Arsenault et al., 2018; Hosseini et al., 2022), it is critical to understand the role of soil properties on model sensitivity.

Here we focus on the response of the NOAH-MP LSM specifically, without an atmospheric model component (i.e. WRF), to two different soil data and schemes for calculating soil parameters with the objective of evaluating the model estimation of the land surface fields. Our study, while complementary to Zhang et al. (2023), seeks to expand the discussion by focusing on a region that is typically energy, rather than water, limited, has intensively managed landscapes and is under a very contrasting climate regime. Additionally, we employ an alternative approach to derive model relevant soil parameters, using pedotransfer functions, and incorporate additional data sources for evaluation of the model responses. Critically, we focus on two contrasting years when model differences are likely to be largest.

Due to its maritime climate, Ireland lies in a temperate region with cool temperatures year round and no marked seasonality to precipitation. As a consequence, growing conditions are near optimal, particularly for agricultural or managed grasslands which account for almost 60 % of the total land area. The country has relatively young (<12-15 Kyrs) and heavily managed soils that are very heterogeneous over small spatial scales. In spite of the maritime climate, variations in the dominant soil categories across the country mean that some locations experience periodic/seasonal soil moisture deficits, particularly in the sandy soils located in the south-east of the island and which experience typically drier and sunnier summer periods, relative to the rest of the country.  To the north and west, soils tend to have higher clay contents, which can act as a buffer to prolonged periods of reduced precipitation or become waterlogged during wet periods. The complexity of Ireland's soil landscapes and climatological regime provide new impetus to test the impact of different soil data representations on LSM simulations, particularly within the context of understanding how projected future changes in the frequency and intensity of drought events may spatially impact maritime temperate locations, such as Ireland.

## 2.    Data and Methods

### 2.1    Background context of Ireland

The climate here is predominantly influenced by the moist mid-latitude westerlies that blow off the North Atlantic Ocean, and occasional incursions of cold air masses during winter (Peel et al., 2007). The long-term (1981-2010) average daily maximum temperature is between 18° and 20°C in summer and 8 °C in winter. Occasionally, the daily mean temperature drops below 0 °C in autumn and winter. Rainfall is distributed throughout the year with a mean annual value of 1200 mm. The west of Ireland typically

experiences higher rainfall amounts (1000-1400 mm), and can exceed 2000 mm in upland areas. Conversely, the east experiences lower rainfall amounts, between 750 and 1000 mm. More detailed information on the background climate of Ireland is provided in Walsh (2012). Although these are typical climatic conditions in Ireland, the country is also prone to extreme weather events. For instance, the summer of 2018 was an exceptionally warm and dry period, associated with a weakened jet stream and persistent region of high pressure over north western Europe; it was followed by a return to normal conditions in 2019.

In relation to the general soil information (Figure 1a), the south-east is characterized as having relatively free draining sandy soils; peat soils dominate the mountains, hills and western edge of the country, while limestone-rich soils dominate the midlands and south (Creamer et al., 2014). Among the land use types (Figure 1b), agricultural grassland dominates the total land area in Ireland, accounting for an estimated 59% of the total land use. The temperate climate in combination with fertile soils, provides conditions that are favourable for near year round grass growth, particularly in the coastal margins and along the south coast. However, cooler temperatures and heavy clay (wet) soils limit the grass growing season (early to mid-March) in the uplands, midlands and north of the country (Keane and Collins, 2004).

## 2.2    Model description

Here, we employ the advanced community NOAH-MP land surface model with multi-parameterisation options, with improved representation of physical processes (Chen et al., 1996; Niu et al., 2011). The model can be run in uncoupled mode, with the capacity to simulate land state variables (e.g. soil moisture) and land energy, water and carbon fluxes. It also represents a LSM that is coupled with numerous atmospheric and hydrological models, including the community based Weather Research and Forecasting (WRF) model (Barlage et al., 2015). Due to the potential for selecting and combining multi-physics options, the model has been widely used for a range of different research applications, including natural hazards, drought and wildfire monitoring, land-atmosphere interactions, sensitivity and uncertainty quantification, biogeochemical processes, water dynamics, dynamic crop growth modeling, and soil hydrothermal processes (e.g. Zhuo et al., 2019; Kumar et al., 2020; Chang et al., 2022; Hosseini et al., 2022; Nie et al., 2022; Warrach-Sagi et al., 2022; Hu et al., 2023).

In NOAH-MP, the major improvements in mechanisms relevant to soil processes are (1) ability to distinguish less and more permeable frozen soil fractions; (2) the introduction of an alternative lower boundary soil temperature that is based on zero heat flux from the deep soil bottom; (3) the addition of TOPMODEL and SIMGM models for runoff and groundwater physics options (Niu et al., 2007); and, (4) the inclusion of

an unconfined aquifer beneath the 2 m bottom of the soil layer to account for water transport between the soil and aquifer. Similar to other LSMs, the NOAH-MP model framework is typical in its ability to define soil properties either by using the dominant soil texture class (e.g. USDA), linked to laboratory- or empirically- derived soil parameter values, or using soil texture (proportions) in combination with PTFs (e.g. Saxton and Rawls, 2006). Of these, the former is most commonly employed, in combination with readily available global soil information.

The prognostic equations from Mahrt and Pan (1984) are used to describe soil moisture and soil temperature in the model (Chen et al., 1996).

$$\frac{\partial \theta}{\partial t} = \frac{\partial}{\partial z}\left(D\,\frac{\partial \theta}{\partial z}\right) + \frac{\partial K}{\partial z} + F_\theta, \qquad (1)$$

$$C(\theta)\frac{\partial T}{\partial t} = \frac{\partial}{\partial z}\left(K_t(\theta)\frac{\partial T}{\partial z}\right), \qquad (2)$$

where $\theta$ is the soil moisture, $C$ is the volumetric heat capacity, $T$ is the soil temperature, and $K$ and $K_t$ are the hydraulic and thermal conductivities, respectively. $D$ is the soil diffusivity and $F_\theta$ are the sinks and sources of soil water, that is, evaporation and precipitation. $C$, $D$, K and $K_t$ are functions of soil texture and soil moisture.

## 2.3    Gridded data

Meteorological variables required as initial and forcing conditions were obtained from the European Centre for Medium-Range Weather Forecasting (ECMWF) database. We employ the state-of-the-art ECMWF ERA5-Land global reanalysis product that provides data at $0.1^\circ$ (~9 km) spatial and hourly temporal resolution (Muñoz-Sabater, 2021). The required forcing variables include total precipitation, incident shortwave and longwave radiation, 2m air temperature, 10m zonal and meridional wind components, surface pressure and specific humidity. For initialisation, the model also requires initial values of soil temperature, surface skin temperature, canopy water and snow water equivalent to be specified for the first timestep. The hourly data for all variables was obtained for the period 2009-2022.

The NOAH-MP model also requires geographical data (e.g. soil texture and land use) and time varying vegetation products (e.g., leaf area index and fraction of green vegetation). We use the STATSGO gridded soil categories map provided at 5 arcmin resolution (~6 km at $52^\circ$N) (FAO 2003a; b) and the International Soil Reference and Information Centre (ISRIC) global SOILGRIDS data (Hengl et al., 2017; Poggio et al., 2021). The latter is available at 250 m resolution and six standard soil depths, however, sand and clay proportions are currently available at four depth layers as part of the WRF geographical data fields. Preprocessing of the data was undertaken in the WRF Preprocessing System (WPS) (Skamarock et al., 2019).

## 2.4   Model simulations

We employed the offline version of the NOAH-MP model (version 4.3) within the framework of the High Resolution Land Data Assimilation System (HRLDAS) (Chen et al., 2007). Using the WPS system, the model domain is set up with a 1 km grid covering the island of Ireland and includes the west coast of the United Kingdom (Figure 1). We incorporated a high resolution land use dataset based on the 100 m raster CORINE Land Cover for 2018 (CLC 2018). The 44 CORINE land cover classes were initially reclassified into 20 categories to match the default modified IGBP MODIS 20-category land use (Figure 1b). The data is then resampled to 250 m using a majority rule. To generate the required geographic files for input to NOAH-MP, the CLC 2018 was converted to binary format which is then used as input to the WPS, which generates the gridded geographic format required to run the NOAH-MP model. Other geographical data, such as topography, green vegetation fraction and surface albedo used in this study are derived from the model default datasets provided by the Research Application Laboratory, National Center for Atmospheric Research (RAL/NCAR).

To investigate the effect of soil hydrophysical properties on model estimated soil moisture and soil temperature, we configure two experiments that are based on different soil data options, namely, (1) dominant soil texture categories used as default in WRF/NOAH-MP; and, (2) soil texture properties (e.g. sand, silt, clay) in combination with PTFs (PTFs based on Saxton and Rawls, 2006). The dominant soil texture option uses the baseline FAO/STATSGO dataset with the empirically-derived soil properties obtained from the model look-up table, while the PTF-derived soil properties use the fine-scale SOILGRIDS sand and clay proportions as input to the PTF equations. The dominant top soils across the domain are broadly classified into four and two categories based on STATSGO and SOILGRIDS, respectively (Figure 2). While Loam and Sandy Loam soil textures cover the largest area in both data sources (Table 2), the extent to which difference in the soil data (e.g. spatial extent of textural classes; soil hydrophysical parameters) contribute to model uncertainty in the NOAH-MP model is evaluated. Other NOAH-MP physics options used are outlined in Table 3.

For the numerical experiments, soil layer thicknesses of 0.07, 0.21, 0.72 and 1.55 m are used, with a cumulative soil depth of 2.55 m. The thicknesses are selected to match the layers of initial soil input fields from ERA5-Land to minimize the effects of interpolation of the boundary data inputs on the model simulation. The model is spun-up over 10 years for each experiment using the climatology of the hourly ERA5-Land for the period 2009-2022, to bring the soils to thermal and hydrologic equilibrium with

the atmosphere. We employ a climatology, rather than preceding meteorology (e.g. 2000-2009), to limit the impacts of unusual or extreme weather events on the estimation of the model stores. After spin-up, the model stores are assumed to be stable and are used as input to initialise the simulations, reported on here, using the hourly meteorological forcing from 2009 to 2022.

2.5    Station data

Profile measurements of soil temperature and volumetric water content (VWC) are obtained from two established eddy covariance flux sites located over grass land cover at Johnstown Castle and Dripsey (Kiely et al., 2018; Murphy et al., 2022), located in the south of the island. In addition, we employed five new sites (deployed as part of a new national network of monitoring sites – Terrain-AI) co-located with existing national meteorological sites, namely Athenry, Ballyhaise, Claremorris, Dunsany and Valentia, and which are distributed across the island (Figure 1a).

The selected sites are characterized as having either loam or sandy loam soils (Table 1), representative of the top two dominant soil texture categories in STATSGO and SOILGRIDS (Table 2); and have contrasting soil water regimes (Figure 1 a). For example, Johnstown Castle is characterized as having imperfectly drained sandy loam soils and a measured field capacity of 0.32; Dripsey is classified as having loam soil and has a measured field capacity of 0.42 (e.g. Peichl et al., 2012; Kiely et al., 2018; Ishola et al., 2020; Murphy et al., 2022), it is classed as poorly drained as it is dominated by heavy soils that retain water throughout the year.

For note,  the flux sites' VWC values are measured in the top 20 cm soil layer, while the Terrain-AI sites measure at fixed depths down the soil profile (e.g. 5 cm, 10 cm, 20 cm, 30 cm, 40 cm, 50 cm, 60 cm, 75 cm and 100 cm). The Terrain-AI network is part of a wider recent national initiative to establish a long-term network of soil moisture monitoring sites across Ireland. It measures in situ soil moisture content using a Time Domain Reflectometry (TDR) profile sensor (Campbell Scientific CS615/CS616). Given that the Terrain-AI sites are relatively new, starting from 2022, the VWC measurements used here are limited to a year, and may be prone to outliers as the TDR probes require some time for the soil to settle around the sensor. However, there is no evidence of TDR sensor decay in the measured VWC when the 2022 values are compared with the patterns found in the more recent data (2023-present) at the 5 cm and 20 cm soil depths (Figure A1). Soil temperature measurements recorded at 5, 10 and 20 cm depths were obtained from Met Éireann, the national meteorological agency, for the same sites as the soil moisture measurements.

Half-hourly or hourly measurements are available for the period from 2009 to 2012 from Dripsey; 2018 (measurements available from the second half of year), 2019 and

2021 from Johnstown Castle, and the year 2022 for the Terrain-AI/meteorological sites – representing different measurements periods and hence data availability at the sites. Metadata for each station, outlining soil type, land cover and altitude are provided in Table 1.

## 2.6    Satellite products

Global satellite soil moisture datasets (e.g. ESA-CCI, SMAP, SMOS, and ASCAT) are often used to evaluate LSM at large spatial scales. Many of these products differ in terms of the satellite sensors and start of operations, and are subject to data gaps, cloud coverage, coarse resolution and limited time coverage (Beck et al., 2021). We employ the Soil Water Index (SWI) product (soil moisture expressed in percentage degree of saturation), derived from the fusion of Sentinel-1 C-SAR (1 km) and Metop ASCAT (25 km) sensors, to evaluate the NOAH-MP model at grid scales (Bauer-Marschallinger et al., 2018). The product is derived from the ASCAT surface soil moisture (SSM) data using a two-layer water balance model that estimates the surface and profile soil moisture as a function of time (Wagner, 1999; Albergel et al., 2008). The operational ASCAT SWI are provided at eight different time characteristics (taken as soil depths), 1km resolution and daily mean values, from 2015 to 2022. The product is archived by the Copernicus Land Service and has been validated in previous studies (e.g. Albergel et al., 2012; Paulik et al., 2014; Beck et al., 2021).

To evaluate our model at grid scales, we employ the characteristic time length T2, representative of the near-surface (0-10 cm), and T10, representative of the subsurface (10-30 cm), soil layers. We choose the ASCAT 1km SWI as the reference satellite product as it provides data at different depth layers, matches the NOAH-MP model grid resolution (e.g. 1 km) and has been found to out-perform other similar products, such as the ESA-CCI SSM and physics-informed machine learning GSSM 1km product (Han et al., 2023), when evaluated against available ground measurements (Figures A2-A3).

## 2.7    Analysis

### 2.7.1   Model evaluation using in situ data

The half-hourly or hourly station data and model outputs for each grid cell are aggregated to daily averages for consistency. For each validation site, variable and available time period, the daily mean values from the respective model grid cell are extracted at the model resolution (1 km). The daily values of topsoil temperature (0-7 cm) and topsoil and sub-surface (7-28 cm) volumetric water content are compared against the available in situ measurements. The model estimated values are then evaluated using the Root Mean Square Deviation (RMSD), Percent Bias (PBIAS) and

Pearson's Correlation Coefficient (R).

*2.7.2   Model evaluation using satellite data*

Given the limited number of in situ sites and scale differences between point observations and model grid resolution, the general interpretation of model performance across landscapes should be treated with care. However, the use of satellite data is a standard and pragmatic way of evaluating model outputs of soil moisture over large spatial scales (He et al., 2023), notwithstanding the inherent uncertainty (e.g. coarse resolution and data gaps) of the satellite products. We evaluate NOAH-MP estimated soil moisture against the ASCAT SWI (Figures A2-A3), for the surface and subsurface layers. To ensure that the NOAH-MP soil moisture is comparable with the ASCAT SWI at the grid scale, we derive a standardized Relative Soil Moisture (RSM) index, which varies between 0 for wilting point and 1 for saturation (e.g. Samaniego et al., 2018), as follows:

$$RSM_{i,j,k} = \left( \frac{\theta_{i,j,k} - \theta_{wilt_{i,j}}}{\theta_{sat_{i,j}} - \theta_{wilt_{i,j}}} \right) x100 \qquad \qquad 3,$$

Where $\theta_{i,j,k}$ is the simulated volumetric water content, $\theta_{sat}$ and $\theta_{wilt}$ are the soil moisture at saturation and wilting point, respectively (Figure 3). We obtain RSM values for both the surface and subsurface soil layers. For the surface layer, ASCAT SWI-002 data, which imply surface soil moisture conditions, are compared against the model derived RSM values for the topmost model soil layer (0-7 cm). For the subsurface, RSM values are taken as the mean aggregated values over the topmost three model soil layers, and are evaluated against the ASCAT SWI-100. Similar metrics are used for the point-scale evaluation (see Section 2.7.1) and are also calculated at grid scale between the reference datasets and model outputs for selected dry (2018) and normal (2019) years.

Additionally, differences between the near-surface soil moisture simulations are quantified for each grid (i,j) using the standard deviation difference ($\Delta\sigma$) as a measure of spread between the two soil datasets.

$$\Delta\sigma_{i,j} = \left[ \sqrt{\frac{\sum_{k=1}^{n} \left(\theta_{i,j,k} - \underline{\theta}_{i,j,k}\right)^2}{n}} \right]_{STATSGO} - \left[ \sqrt{\frac{\sum_{k=1}^{n} \left(\theta_{i,j,k} - \underline{\theta}_{i,j,k}\right)^2}{n}} \right]_{SOILGRIDS} \qquad 4,$$

where, $\theta$ is the VWC value at time k and n is the total number of daily soil moisture values from 2009-2022.

*2.7.3   Transition from energy limited to water limited regime*

We also analyse the potential of NOAH-MP for simulating the evolution of an agricultural drought across the domain. Since the west-central European summer drought of 2018 was an exceptional event in terms of hydrological extremes across Ireland (Met Éireann Report, 2018; Falzoi et al., 2019; Moore, 2020; Ishola et al., 2022), we evaluated the model over this period. We apply grid-scale cumulative RSM values integrated over the three topmost soil layers (0-100 cm) (Section 2.7.2), due to its simplicity and ease in quantifying and interpreting available soil water. Additionally, the RSM metric reduces the impact of systematic biases in absolute values and/or the impact of transient errors associated with short-term fluctuations in absolute VWC values. In principle, RSM is an important drought indicator, particularly at short-time scales, and analogous to the widely used Soil Moisture Index (SMI) for drought monitoring (Samaniego et al., 2018; Grillakis, 2019). To characterise decreasing soil moisture during a drought period, percentiles of RSM values per grid cell are calculated based on 7-day moving windows from June to August for the climatology period 2009 - 2022. This amounts to 98 samples (7 days x 14 years) as input per window. For individual model experiments, STATSGO and SOILGRIDS, the derived spatial RSM percentiles per day in each window are then classified into different drought categories ranging from least to most severe (Table 5), following Xia et al. (2014). These categories are currently employed by the U.S. Drought Monitor (USDM) for operational and regionally specific drought monitoring (Svoboda et al., 2002).

**3.    Results**

First, we present a comparison of the ERA5-Land total annual precipitation against station data, to identify any significant differences between the observed and input meteorology, for the respective measurement periods. Figure 4 shows that the total cumulative precipitation over the periods of interest are well replicated in the ERA5-Land precipitation data across the selected stations, including for the extended period of no rainfall during the summer months of 2018 (Figure 4 f).

3.1    Model evaluation: Soil moisture

*Station observations*

The results of model simulations of near-surface and subsurface volumetric water content (VWC in $m^3$ $m^{-3}$) for both STATSGO and SOILGRIDS are presented for the periods when measurements are available at the selected sites. Figures 5 and A4 illustrate the comparisons and error statistics of near-surface VWC between the measured (0-5 cm) and modelled (0-7 cm) layers, while the subsurface VWC is illustrated in Figure A5. It is important to note that we are comparing a 1 km model grid (areal) to a measurement point, which are assumed to be equivalent. Also, we are

evaluating the model simulations within the top 20 cm VWC values at Johnstown Castle and Dripsey, the two flux sites, in the absence of near-surface (0-5 cm) VWC data for these locations.

Based on the analysis, the near-surface simulations are in closer agreement with the observed VWC at Athenry, Claremorris and Johnstown Castle with the lowest error statistics (RMSD ≈ 0.1 $m^3$ $m^{-3}$, PBIAS < ~25%) relative to other stations (Figure A4). While the model outputs appear to more closely match the observations during the summer months at Valentia (Figure 5e), the model significantly underestimates the measured VWC outside of these months, impacting the overall model performance at the station (Figure A4). The Pearson's correlation is generally high, above 0.8, across the measurement sites, with the exception of Ballyhaise (>0.71) and Claremorris (>0.63). The lowest model performance in terms of RMSD and PBIAS occurs at Dunsany, Valentia and Dripsey, with RMSD > 0.15 $m^3$ $m^{-3}$, PBIAS > 30% (Figure A4). Model simulations with both soil datasets broadly underestimate the observed VWC values in the autumn and winter months, but the model bias is lower in the STATSGO experiment compared to SOILGRIDS, a finding that is broadly consistent across the stations (Figure A4). Dry biases (0.15 - 0.4 $m^3$ $m^{-3}$) are evident in autumn and winter during which the measured VWC values are higher (Figure 5 a-e), except at Dripsey where a systematic dry bias is evident throughout the entire simulation period (Figure 5g). Conversely, during summer when soil moisture conditions tend to dry in response to atmospheric forcing (e.g. higher global solar radiation and evaporation), VWC temporal patterns are reasonably captured by both model experiments (biases are less than 0.1 $m^3$ $m^{-3}$), including during 2018, which experienced exceptionally dry soil moisture contents during the summer months (Figure 5f). The differences between STATSGO and SOILGRIDS are relatively small (< 0.05 $m^3$ $m^{-3}$) across the year(s); but seasonal differences are evident at some sites, likely due to the generally higher soil porosity and FC values in STATSGO relative to SOILGRIDS (Figure 3 a-f).

Interestingly, both model experiments are capable of broadly replicating the measured near-surface VWC values at Athenry (well-drained), Claremorris (well-drained) and Johnstown Castle (imperfectly drained), where the soils are classified as either well- or imperfectly- drained (Figure 1a; Table 1), but the simulations underestimate the variability (Figure 5 a, c, f). In contrast, for locations classified as poorly drained, namely Ballyhaise, Dunsany and Dripsey (Figure 5 b, d, g), the model does not perform well. The model appears to be able to replicate measured VWC during the summer months at Valentia, which is classified as well drained, but performs poorly for the

remaining months (Figure 5 e). Figure 5 (boxplot) further illustrates the summary statistics and spread in the model simulated and observed VWC. The mean observed VWC ($\approx$0.3 m$^3$ m$^{-3}$), calculated over the available measurement periods, is better captured in STATSGO than SOILGRIDS, particularly at Athenry, Ballyhaise, Claremorris and Johnstown Castle. However, where the mean observed VWC exceeds this value (e.g. > $\approx$0.3 m$^3$ m$^{-3}$), both experiments lead to significant underestimation of VWC, as evident at Dunsany, Valentia and Dripsey.

*Model comparison with reference ASCAT satellite SWI data*

While the selected measurement stations are well distributed and represent different soil moisture regimes across Ireland (Figure 1a), given the relatively small number of stations, generalising the results to the entire domain may not be justified. To address this, we evaluated all model grid cells individually against the reference ASCAT satellite data. Prior to undertaking the grid based analysis, we compared the ASCAT SWI, rescaled to match the mean and standard deviation of the measured values at the site of interest, to the available measured data at the sites. The ESA CCI SM is also included in the figures, however, the ESA CCI SM product reports absolute values of VWC (m$^{-3}$ m$^{-3}$) for the top layer and is at 0.25$^\circ$ resolution. On the basis of the rescaled values, the ASCAT SWI is shown to largely reproduce the temporal variability of the measured values indicating its suitability for use across the domain (Figures A2-A3). Figure 6 shows the results of the all island grid-scale evaluation (n = 131,000 grid values), which compares daily RSM values, derived from the STATSGO and SOILGRIDS simulations, against the reference ASCAT SWI at the surface and subsurface for the 2018 dry and 2019 normal years. Median metrics for each soil texture category in STATSGO and SOILGRIDS are presented in Tables 5 and 6.

As shown in Figure 6 (top) for the 2018 dry year, the median statistics indicate that STATSGO has lower RMSD values compared to SOILGRIDS for both the surface and subsurface layers and PBIAS values that lie closer to 0. While the Pearson's R statistic (median around 0.85) for STATSGO and SOILGRIDS is comparable for the surface layer, the SOILGRIDS experiment produces a higher R value in the subsurface layer during the dry year. For the 2019 normal year (Figure 6, bottom), SOILGRIDS displays equivalent or lower error statistics for the surface layer, with a median RMSD of 0.016 %, PBIAS of around 1 % (6 % for STATSGO) and R of 0.73. For the subsurface layer, SOILGRIDS produces better results than STATSGO with lower RMSD (0.01 %) and PBIAS (6%) and a higher R value (median approx. 0.76).

The extended tails (positive/negative in PBIAS and lower/higher in RMSD and R) in the density distribution indicate the spread in RMSD, PBIAS and R values. Given that the Loam (L) and Sandy Loam (SL) soils represent the largest proportion of grid cells across the study domain and are relatively comparable in terms of spatial coverage in STATSGO and SOILGRIDS (Table 2), the error statistics for these soil texture categories are further explored here. For 2018, results show that both experiments produce lower RMSD error statistics for SL than L at the surface layer, while STATSGO has lower PBIAS for SL than L (Table 5). For the subsurface layer, both soil datasets have similar RMSDs and have lower PBIAS for L, compared to SL. For the 2019 normal year (Table 6), both STATSGO and SOILGRIDS show improved PBIAS for L, compared to SL, in both the surface and subsurface layers. STATSGO has equivalent or lower RMSD and lower PBIAS error statistics than SOILGRIDS at the surface layer. The RMSD and R statistics are relatively comparable in both the surface and the subsurface layer for both the STATSGO and SOILGRIDS simulations and for L and SL soil categories. However, STATSGO produces lower PBIAS statistics than SOILGRIDS for SL in 2018 (surface and subsurface) and SL (surface) and L (surface and subsurface) soil in 2019. For 2019, these findings contrast with those of the previous analysis, based on all grid cells and independent of soil texture class (Figure 6).

The spatial characteristics of the ASCAT SWI and model derived surface RSM values are shown in Figure 7 a-j, along with their difference, for the years 2018 and 2019. The long-term seasonal differences in the surface VWC between both experiments are also shown in Figures A7-A8. For the surface VWC, both simulations largely exhibit a dry bias, increasing from the north west to the south east of the country; higher biases are evident in the eastern and southern parts of the country in SOILGRIDS relative to STATSGO (Figure 7). The higher (dry) biases in both STATSGO and SOILGRIDS occur in regions that are largely classified as L soil texture class in both soil datasets. The dry bias is larger in 2019, compared to 2018 (dry year) and higher for SOILGRIDS than STATSGO. For the subsurface values (Figure A6), wet biases are evident in the north west, west and south west, which are characterised as SL and Clay Loam in STATSGO and SL in SOILGRIDS. Towards the south and southeast of the domain, the results shift towards a dry bias, mostly in areas represented by L soils; more spatially extensive wet biases are evident in the normal year 2019, compared to 2018. While the  spatial patterns in the wet and dry biases are broadly consistent for both experiments and years, the dry bias in both years is more pronounced in SOILGRIDS than STATSGO, consistent with the surface layer. Conversely, the wet bias in the sub-surface layer is more widespread in STATSGO than SOILGRIDS. While both soil datasets show the largest difference between the modelled and ASCAT SWI surface

layers in the south eastern part of the country, this region displays the smallest between model differences (< 0.05 $m^{-3}$ $m^{-3}$) on a seasonal basis (Figure A7). As expected, the largest differences between the model estimated VWC are located in regions where the soil datasets have different soil texture classes (Figure 2 c) and hence associated soil properties. For example, STATSGO has a region of clay loam (CL) soils to the north west and clay (C) soils on the west coast, in contrast to the SOILGRIDS L class, and have different soil properties associated with these classes (Figure 3); the largest differences between the model runs (STATSGO – SOILGRIDS) are associated with the STATSGO clay loam locations, with STATSGO indicating generally wetter soils associated with both the clay loam and clay texture classes. While the wilting points are similar between both datasets, STATSGO has higher field capacity and soil porosity for these textural classes (C, CL) (Figure 3). Both soil datasets have SL classes located along the western seaboard, however, STATSGO estimates lower VWC compared to SOILGRIDS in these regions (Figure A7).

## 3.2    Model evaluation: Soil temperature

Figure 8 (a-g) illustrates model comparisons against the reference station measurements of topsoil (0-5 cm) temperature, while Figure A9 shows the associated evaluation results. Generally, the error statistics (RMSD and PBIAS) for both the STATSGO and SOILGRIDS experiments are low, and R values are high (above 0.9 across all sites). The model is closer to the observations in Athenry, Dunsany, Valentia and Johnstown Castle (RMSD < 3 K and PBIAS < 1%), compared to Ballyhaise, Claremorris and Dripsey where the errors exceeded these values. Comparatively, SOILGRIDS leads to a slightly better model performance than STATSGO across the sites.

The spread of the observed soil temperatures are reasonably replicated in both experiments and for the selected year(s) across locations (Figure 8, bottom). Whereas the mean of the observed soil temperature, which is approximately 285 K, is systematically underestimated by between 1 K to 3 K across stations; however, the peak values in the mid-summer months are well captured by both experiments (Figure 8a-g). Overall, both STATSGO and SOILGRIDS produce covarying soil temperature profiles, but the differences between the measured and simulated values are statistically significant (*p-value* < 2.2 x $10^{-16}$) for all sites.

Given the reasonable model performance across the selected locations, the grid-scale model differences in soil temperature between STATSGO and SOILGRIDS is further examined (Figure 9). The spatial differences of surface soil temperature are based on the seasonal climatology from 2009 to 2022. In response to seasonal variations in global solar radiation and VWC, winter shows the lowest soil temperatures (Figure 9

a,e), whereas summer is characterised as having the highest soil temperatures (Figure 9 c,g), widespread mostly over Loam soil in the south and southeast of the study domain. The south and east are seasonally drier, experiencing lower rainfall and soil water deficits during the summer months (Figures 1a and A7). The spatiotemporal evolution of the soil temperature characteristics is consistent in both STATSGO and SOILGRIDS throughout the year. Both soil datasets produce soil temperature differences that are low or negligible in the south and southeast, which are dominated by Loam soils (Figure 9 i-l). However, STATSGO exhibits colder soil temperature in Clay and Clay Loam soils, and warmer Sandy Loam soils in the north and southwest, with respect to SOILGRIDS. These areas exhibiting cold and warm soil temperature differences between STATSGO and SOILGRIDS, coincide with regions exhibiting wet and dry VWC biases. (Figure A7).

3.3     Spatial and temporal evolution of soil moisture drought

Figure 10 illustrates the spatial characteristics of 0-100 cm RSM percentiles for selected days during the summer of 2018. The selected dates denote the start, peak and end of the summer water deficits (Figure 4 f) experienced during that year.  For the first 7-day window ending 07 June, the southeast and east of Ireland show low drought intensity D0-D1 (abnormal/moderate) in STATSGO, compared to SOILGRIDS which exhibits values in the severe drought D2 category. During this build up period, there are notable spatial differences between STATSGO and SOILGRIDS, with the latter exhibiting a more spatially extensive region in the D0 and D1 categories.

By the middle of summer 2018 (sixth week ending 12 July), almost the entire island is dominated by the exceptional drought D4 category in STATSGO, except for areas in the extreme north east and south west which are classified in the D2 and D3 categories. These patterns are broadly consistent in SOILGRIDS except for small areas in higher intensity drought classes. For example, the drought category in the north east of the island shifts from D2 in STATSGO to D3-D4 (extreme and exceptional) categories, and from D2-D3 (severe and extreme) category in the southwest and east of Ireland to D3-D4 drought categories in SOILGRIDS. It is notable that these regions in the southwest and east are associated with high topography.

Whereas the soil water deficits appear to have improved by the end of summer (week 13 ending 30 August), the landscape is experiencing different levels of soil water deficits. For example, in STATSGO, the moderate drought D1 category broadly dominates the Loam areas in the midlands, south and southeast of Ireland, while a mix of drought D1-D4 categories dominates the west and southwest of the country. These patterns are consistent in SOILGRIDS, but D3-D4 drought categories remain more extensive in the north, west and southwest in SOILGRIDS compared to STATSGO.

Figure 11 illustrates the time-areal coverage of the drought categories over the domain during the summer period 2018, based on RSM percentiles. While the landscapes are already experiencing soil water deficits by the start of June, the largest areal coverage (about 70 % in STATSGO and 80 % in SOILGRIDS) is dominated by low drought intensities (D0-D2). Approximately 10 % of the domain is characterised by extreme and exceptional D3-D4 drought, up to the end of June. The drought intensifies effectively from late June, with higher areal coverage evident in the D4 category (more than 80 %), extending for several days in STATSGO (July 10-15). Over the same period, the D4 category in SOILGRIDS is less extensive and lasts for a shorter period than STASGO, but also transitions to less severe categories more slowly than STATSGO. At the start of August, there is a brief interlude with a reduction in the areal extent of the high intensity D3-D4 drought evident in both SOILGRIDS and STASGO, which transition to the less severe categories D0-D2. By the last week of August, the peak of the drought has passed and the landscape begins to recover.

## 4.    Discussion

4.1 Effects of soil hydrophysical properties on simulated soil hydrothermal regimes.

In this study, we investigated the differences between two global soil texture data sets currently implemented in the NOAH-MP land surface model on the simulated soil hydrothermal properties. In addition to using the default look-up table in combination with the STATSGO soil information, which is perhaps the most widely used or typical approach, we employed PedoTransfer Functions (PTFs) in combination with the SOILGRIDS soil information to explore the impact of different soil datasets and hence their associated soil properties (e.g. porosity, field capacity, wilting point, hydraulic conductivity, etc.) on the simulated surface and subsurface soil hydrothermal parameters, during a normal (2019) and extremely dry (2018) year. The role of these properties, particularly the field capacity – a measure of water retained in the soil at the pressure of -0.33 bar after excess rainwater has drained off - are critical to correctly simulating soil hydrophysical processes and have consequent impacts on the subsequent interactions between the land surface and the overlying atmosphere (Dennis and Berbery, 2021;2022; Zhang et al., 2023; Zheng and Yang, 2016).

Initially, we compared the model simulated values at grid scale with available in-situ data for a selection of sites distributed across the island and representative of the dominant soil textural properties (Table 1). In general, both the STATSGO and SOILGRIDS model simulations resulted in an underestimation in the modelled variance at all sites compared to the measured values. With the exception of STATSGO at Ballyhaise, both model simulations underestimated the mean observed

values, particularly marked at three sites; seasonal differences were also evident (Figure 5). With the exception of Valentia, SOILGRIDS estimated lower mean values, compared to STATSGO (Figure 5h). At two sites, Ballyhaise and Dunsany, both soil datasets resulted in an overestimation of VWC during the drier summer months, when the measured values indicate the soils were close to, or at, wilting point. The largest differences between the modelled and measured VWC occurred at sites where the soils appear to have a larger water holding capacity, namely Dunsany, Valentia and Dripsey (Figure 5 boxplot). Despite the misrepresentation of the soil texture class and the difference in soil depths between the measured and simulated VWC at Johnstown Castle (Table 1), the model performs reasonably well at this site. However, for a relatively wet site (e.g. Dripsey) where the soil textural class is correctly represented in both soil databases, the model simulations systematically underestimate soil moisture content (Figures 5g and A4). This suggests that the soil-induced model uncertainty which is often linked to misrepresentation of soil texture class (e.g. Zheng and Yang, 2016), and hence misspecification of hydrophysical parameters, can arise due to other factors (e.g. model physics, incorrect hydrophysical parameters etc).

We also compared the ASCAT SWI with the measured VWC at the selected sites and subsequently the RSM derived from the model simulated VWC. Based on the rescaled SWI, derived using the mean and standard deviation of the measured values, the ASCAT SWI is shown to largely replicate the temporal variability of the measured values at the selected sites, in particular the seasonal evolution of soil moisture. With regards to the comparison between ASCAT SWI and the model derived RSM, we found that while the median correlation between SWI and RSM was higher for SOILGRIDS than STATSGO for both the surface and subsurface layers, STATSGO performed better in terms of the error statistics in the dry year (2018), while SOILGRIDS performed better in the normal year (2019) (Figure 6). While both the SWI and RSM are based on relative, rather than absolute values, the calculated correlation coefficients (R values) indicate that the model is able to capture at least some of the temporal evolution (covariation) of soil moisture in both a dry (2018) and normal year (2019) and importantly, suggests that the model soil physics is functioning correctly or at least in a way that is temporally consistent with the independently derived ASCAT SWI data. However, while both STATSGO and SOILGRIDS produce similar estimates of VWC where textural classes are in common (Figure A7), both STATSGO and SOILGRIDS systematically under estimate VWC, when compared to the ASCAT SWI, and in particular for the Loam textural class (Figure 2; Figure 7); SOILGIRIDS shows a larger underestimation compared to STATSGO (Figure 7; Figure A7) most marked in winter, spring and autumn (Figure A7). From Figure 3,

STATSGO has higher field capacity and wilting point values associated with Loam soils, compared to SOILGRIDS, which may explain the lower bias in STATSGO, relative to SOILGRIDS.

The assessment of the model against the measured values (Figure 5) and the ASCAT SWI (Figure 6; Figure 7) highlight the potential impact of the prescribed soil hydrophysical parameters, specifically FC and WP, in limiting the models ability to accurately simulate absolute values of soil moisture content within the model soil layers. To test this, we focus on two sites for which measured FC is available, namely Johnstown Castle and Dripsey. The measured field capacity (FC) in the top 20 cm at Johnstown Castle is 0.32 $m^3$ $m^{-3}$ (Table 1) (Peichl et al., 2012), which lies close to the representative FC value employed in both STATSGO and SOILGRIDS for this location. However, the measured FC in the top 20 cm at Dripsey is 0.42 $m^3$ $m^{-3}$ (Table 1), higher than the respective FC value of ~0.31 $m^3$ $m^{-3}$, prescribed from STATSGO, via the lookup table, and the value from SOILGRIDS using the PTFs, for this location (Figure 3 and 5 boxplot). While the model estimated VWC at Johnstown Castle lies close to the measured values at this site, the model systematically underestimates VWC at Dripsey. Ultimately, a lower FC limits the ability of the soil to increase the memory of the stores, resulting in a systematic bias in the simulated VWC. To illustrate the role of the prescribed FC value at Dripsey, the simulated VWC for a neighboring grid cell with a FC of 0.412 $m^3$ $m^{-3}$ and which experiences similar weather conditions is plotted against the measured VWC at Dripsey (Figure 12). A higher FC clearly results in higher VWC values, significantly reducing the systematic bias (RMSD and PBIAS) between observations and STATSGO by more than 50 % of the FC value employed by the model at Dripsey. In contrast, the maximum FC derived from SOILGRIDS across the domain is 0.34 $m^3$ $m^{-3}$ (Figure 3), which lies around the default value, and is not in a proximal grid location to the Dripsey site. Hence, using the same grid cell as above, SOILGRIDS with PTFs fall short of this and consequently fail to improve the simulated VWC.

While the choice of PTFs is critical in model simulations of soil water fluxes (Weihermüller et al. 2021), the default Saxton and Rawls (2006) PTFs produce properties that lie close to the look-up table in NOAH-MP model. One reason for this similarity is that in general the SOILGRIDS sand and clay compositions produce a similar spatial distribution in the Loam and Sandy Loam soil texture classes that coincide with the locations of the FAO/STATSGO classes (Figure 2 and Table 2). Another reason for similar soil properties between the PTFs and look-up table, is the default PTF coefficients are derived based on USDA soil samples (Saxton and Rawls, 2006) and are therefore not likely to be representative of soil processes and

consequently properties in a different study domain; the empirically-derived look-up table values are also based on soil samples from the US. The net effect of similar but inaccurate soil properties is the significant under-representation of soil hydrothermal regimes in wet soils as illustrated in Figures 5 and 7. This aligns with Vereecken et al. (2010) who demonstrated that PTFs are highly accurate over the areas for which they have been developed, but have limited accuracy if transferred outside these areas. Weber et al. (2024) also noted that the divergence between the scale of derivation from laboratory experimental data, and the regional/global scale of application is a fundamental shortcoming for PTFs.

In situations where the model systematically under- or over- estimates soil moisture, the impacts on the surface exchanges with the atmosphere may be more limited (e.g. Dripsey Figure 5g); however, for locations with a high water table and/or subject to seasonal drying (e.g. Dunsany, Ballyhaise Figure 5 b and d), deficiencies in the model estimated timing and extent of soil moisture deficits are likely to result in large seasonal biases in the simulated surface fluxes. However, further work is required to understand the simulated soil moisture response at these locations, but is likely due to a combination of the hydrothermal parameters.

With regards to the model simulated soil temperature, both the STATSGO and SOILGRIDS inputs were able to reasonably replicate the measured surface soil temperature at the selected sites, albeit with a tendency to systematically underestimate the measured values (Figure 8). Only minor, insignificant, differences were evident between the two simulated soil temperature series. In contrast, spatial differences between the STATSGO and SOILGRIDS data were evident, particularly in the north, west and southwest of Ireland (Figure 9), which are largely coincident with the differences in the spatial distribution and extent of selected hydrothermal parameters, between both datasets (Figure 3). Notably, the STATSGO data represents smaller soil grain sizes in most of these areas, relative to SOILGRIDS. This results in higher values of soil hydrophysical properties in STATSGO, including porosity and field capacity, and lower saturated hydraulic conductivity (Figures 3 and A9). The increasing grain size leads to wetter and colder soils in STATSGO, relative to SOILGRIDS in the top 30 cm layer (Figures A6-A7, 7 and 9). Similar to our results, it has been demonstrated that a reduction in soil grain size (e.g. Loam to Sandy Loam) leads to dry and hot soil differences (decrease in latent heat flux and increase in sensible heat flux) between two global soil datasets (Dennis and Berbery, 2021).

Overall, the results here support previous findings that indicate that soil hydrophysical parameters directly impact the model simulated soil moisture; while the spatial distribution of soil textural classes impact soil thermal properties. In contrast to our expectations, the model estimated VWC values were close to the measured values at Johnstown Castle, a site that experiences seasonal/periodic soil moisture deficits/drought, due to a combination of meteorology and soil type (e.g. imperfectly drained). The model performed poorly with respect to the measured VWC at Valentia, (south west coast – well drained), Ballyhaise (north; poorly drained) and Dunsany (east; moderately drained), but highlight that impacts are likely to be more pronounced in relatively wet sites and sites that experience a marked seasonal contrast in soil moisture – which represents a new contribution to the discussion.

4.2. Sources of uncertainties

Model uncertainty: The NOAH-MP model's reliance on default look-up tables for STATSGO and more sophisticated PTFs for SOILGRIDS, introduces systematic biases, particularly when their parameterisations do not represent the local soil conditions accurately. For instance, a mismatch in FC values at Dripsey significantly underestimates soil's water retention capacity, which directly affects soil moisture, with biases exceeding 50% of the employed FC value. In essence, the mismatch in spatial scale between the parameterisation of soil properties and their application in a global model introduces significant uncertainties in soil moisture simulations, particularly in regions with distinct soil properties (Vereecken et al., 2010; Weber et al., 2024). As a consequence, the impact may directly affect the soil moisture coupling with the atmosphere through surface energy fluxes, leading to uncertainties in surface exchanges.

Soil dataset uncertainty: The magnitude of impact of soil dataset uncertainty is particularly pronounced when it comes to the parameterisation of critical soil hydrophysical parameters like field capacity (FC) and wilting point (WP). As shown in this study (Figure 12), a small difference in FC values (e.g., 0.31 m³/m³ vs 0.42 m³/m³) can significantly alter the simulated volumetric water content (VWC), leading to a systematic bias in the model outputs. At sites like Dripsey, where the field capacity was significantly underestimated, the model consistently underestimated soil moisture. This bias was reduced when using a higher FC value for a neighboring grid cell, demonstrating that even small changes in soil property inputs can have substantial impacts on model outputs. Additionally, regional differences in soil properties, linked to divergence in grain size representation between STATSGO and SOILGRIDS (Figures 2-3), affect simulations by 10-30% depending on soil textural class and

climatic conditions. This is evident in regions with high water tables or areas subject to seasonal drying (Figure A7), where the model's inability to accurately simulate soil moisture deficits may potentially propagates through hydrological and thermal cycles, mischaracterising droughts or waterlogging events and affecting surface energy partitioning and land-atmosphere interactions (Dennis and Berbery, 2021; 2022; Zhang et al., 2023).

Observation uncertainty: This also arises, particularly in terms of spatial variability and accuracy of in-situ measurements used for model evaluation. The precision and accuracy of new Terrain-AI TDR measurements used in this study, depend on the sensor installation and performance (Briciu-Burghina et al., 2022). The Terrain-AI network has followed and used the standard, custom-designed installation and calibration tools recommended by the manufacturers, thus we do not observe sensor decay or random errors in the soil moisture measurements, given that the 2022 pattern is temporally consistent with more recent measurements (Figure A1). The observed standard error in the measurements is generally less than 0.01 $m^3$ $m^{-3}$, which is consistent with the recommended optimal accuracy for TDR sensors (e.g. Blonquist et al., 2005). However, we acknowledge that the presence of air gaps between the soil and sensor contact during installation may introduce errors, particularly at the start of sensor measurement. The time for the soil to properly settle around the sensor depends on soil condition and it's a common error for newly installed soil moisture sensors (Briciu-Burghina et al., 2022). Despite this, we believe the impacts on the overall uncertainties in our model evaluation may be relatively small given the observed sensor accuracy across sites.

The in-situ soil moisture measurements, though accurate, are point-based and may not represent grid-scale heterogeneity. For example, discrepancies between measured and simulated volumetric water content (VWC) at Johnstown Castle and Dripsey highlight this limitation (Figure 5). Differences between the measurement depth (e.g., 5 cm top 20 cm, etc.) and model representation (0-7 cm) exacerbate observational uncertainty. For example, model biases at Valentia and Dripsey partly stem from mismatches in vertical soil layering, with the shallower model soil depth expected to be wetter between rainfall events and drier in response to atmospheric conditions. The point-to-grid biases and soil depth mismatches contribute to about 5-20 % errors in validation results, which can distort the interpretation of model accuracy and reliability.

The use of ASCAT characteristics time length (e.g. T2) to represent soil depths without accounting for soil textural class or properties may also influence the model results, as the optimal characteristic time lengths differ for different soil texture categories (de

Lange et al., 2008). The ASCAT SWI replicates the covariation in the measured soil moisture well (Figures A2-A3), but struggles with accurately predicting the absolute moisture content. The correlation between the model RSM and ASCAT SWI was generally higher for SOILGRIDS compared to STATSGO, particularly in a normal year (2019), whereas STATSGO performed better in the dry year (2018) (Figure 6). This indicates that while the model physics and soil properties are functioning reasonably well in simulating temporal variations, there remain issues with absolute soil moisture content.

Overall, global soil datasets may be relevant for weather and climate modelling, assuming the soil water physics are functioning correctly and that the model simulated soil water changes result in the correct partitioning of energy; however, numerous authors (e.g. Dennis and Berbery, 2021; 2022; Zhang et al., 2023) have found that flux partitioning is negatively impacted by the simulated soil moisture. Also, for operational purposes for estimating soil moisture, more refined national level soil data information should be considered. Such efforts, as previously attempted in studies like Reidy et al. (2016), could be expanded to generate more detailed and region-specific soil property datasets.

4.3 Implications for regional drought monitoring

Soil moisture content is an essential variable in many hydrological applications and in understanding the evolution and characteristics of extreme climate events such as droughts. Instead of heatwaves, the study domain is subject to rainfall extremes (Noone et al., 2017), a precursor of soil water deficits and droughts; the intensity and frequency of which have been projected to increase globally and in the study domain by the end of century (Seneviratne et al., 2012; Fealy et al., 2018).

In this study, the drought analysis is based on the cumulative RSM percentiles aggregated over three uppermost soil layers (0-100 cm) for 2018 summer hydrological extremes for STATSGO and SOILGRIDS (Figures 10-11). The 0-100 cm depth is sufficient for drought assessment since the root zone of many crops grown across the world does not surpass 1.0 m in depth (Fan et al., 2016; Grillakis et al., 2019).

Both STATSGO and SOILGRIDS are largely consistent in terms of the peak of soil moisture drought in space and time. However, SOILGRIDS exhibits higher and wider drought intensity in many areas during the buildup and recovery phases, relative to STATSGO. This suggests that there is sensitivity during the buildup to the drought and rewetting of the soils after peak droughts. Similar results have been found in Zheng and Yang (2016), where regardless of soil type, soils tend to dry up with increasing aridity so that the difference in soil moisture between two soil datasets tends to zero. The higher drought intensity of SOILGRIDS is associated with underrepresented soil

hydrophysical properties and simulated VWC as previously highlighted (Figures 3 and A7).

During the summer of 2018, particularly from late May to late July, Ireland was reported to have experienced different degrees of meteorological droughts (rainfall deficits) (Figure 4f) ranging from dry spells to absolute droughts (Met Éireann Report, 2018; Falzoi et al., 2019; Moore, 2020). Meteorological droughts precede soil moisture/agricultural droughts through reduction in soil water storage and available water for plant uptake, our results indicate that extreme to exceptional soil moisture droughts are only effective from last week in June, covering the large part of the domain by mid-July (Figure 11). During August, rainfall improved soil water stores (Figure 4f) and weakened drought conditions across much of the country, particularly in the north and west (Met Éireann Report, 2018; Moore, 2020).

Overall, the discrepancies between STATSGO and SOILGRIDS impacts drought characteristics mostly in space, with SOILGRIDS shifting the abnormal/moderate/severe droughts in STATSGO to extreme/exceptional droughts. These underscore the sensitivity of soil information on drought events, which are critical to improve our understanding of the consequences on ecosystems with regards to predicting the response and productivity, as drought stress has been highlighted as the primary factor limiting ecosystem response and productivity (De Boeck et al., 2011).

## 5.    Conclusions

In this study, the usability of two global soil datasets for representing soil processes in the NOAH-MP model and simulating soil hydrothermal variations and associated extremes, has been evaluated across all of Ireland. Specifically, FAO/STATSGO dominant soil texture categories linked to an empirically-derived soil hydrophysical properties from a look-up table (default in WRF), are compared with PedoTransfer Functions (PTFs) that ingest an alternative SOILGRIDS sand and clay compositions at four soil layers. Through temporal comparison with in situ soil moisture and soil temperature observations, it has been found that both soil datasets can fairly replicate the general soil hydrothermal variations for stations with moderate spikes. However, they under-represent the soil properties (e.g. field capacity) in wet loam soil, leading to systematic dry bias in soil moisture. The results have further shown that there is no distinct difference between the soil physics applied to the same soil texture category in both STATSGO and SOILGRIDS. But, the disparities and sensitivity to soil physics increase for different soil texture categories between the datasets.

Through spatial comparison with satellite-based ASCAT SWI, sub-surface dry bias is more pronounced and widespread in the midland, south and east in SOILGRIDS, while wet bias dominates the west and north. As a consequence, 2018 summer soil moisture droughts broadly intensify more in SOILGRIDS, indicating higher sensitivity during transition to and from peak drought than in STATSGO. This heightened sensitivity could suggest that SOILGRIDS captures finer details of soil moisture variability, however, the disparities could result in inconsistencies in drought response and increase the risk of over-preparation due to overly sensitive model results. Climate change is expected to drive greater fluctuations in soil wetting and drying in Ireland and other regions. This highlights the importance of addressing inconsistencies between soil datasets, not only to better understand the sensitivity of soil information to drought conditions but also to ensure careful interpretation of soil moisture data. Additionally, adopting ensemble approaches could offer a more balanced perspective. Uncertainties in soil moisture simulations are found to be largely linked to soil properties, particularly the field capacity, wilting point and saturation derived from different soil physics, Overall, the study highlights the shortcomings of global soil databases in simulating soil hydrothermal changes and underscore the need to optimize and improve global soil hydrophysical properties that are ingested in LSMs for better performance. Developing detailed regional soil texture properties may be more realistic and enable more improvement in model simulations. Ultimately, this would advance the understanding of the role of soil processes in hydrologic cycle, ecosystem productivity, drought evolution, land-atmosphere interactions and regional climate.

A number of initiatives (e.g. Terrain-AI) has been developed to deploy soil moisture measuring networks across Ireland to address the lack of soil moisture observations. A significant conclusion of this study is that the NOAH-MP model has shown an excellent capacity to ingest better alternative soil texture data, to reduce the model biases of soil hydrothermal changes and evolution of soil moisture drought. Therefore, it can be applied to augment the current network of sites across the country for operational modeling and real-time forecasting of soil moisture conditions and drought across the domain. This will support hydrometeorological monitoring similar to Global Food Awareness System (GloFAS) and NASA's Short-term Prediction Research and Transition with Land Information System (SPoRT-LIS).

**Code and data availability**

The open-source HRLDAS/NOAH-MP model is freely available on github (https://github.com/NCAR/hrldas). The ERA5-Land hourly input meteorological forcing were downloaded from the climate data store (https://cds.climate.copernicus.eu/). The WPS

geographical data were downloaded from NCAR (https://ral.ucar.edu/model/noah-multiparameterization-land-surface-model-noah-mp-lsm). 2018 Corine land use and satellite ASCAT soil water index are freely available on Copernicus Global Land Service (https://land.copernicus.eu/global/index.html). In situ data for the selected sites were obtained from Met Eireann, Ireland and from the European fluxes database cluster (http://www.europe-fluxdata.eu).

## Competing interests

The contact author has declared that none of the authors has any competing interests.

## Acknowledgments

We thank Gary Lanigan for granting access to measurements from Johnstown Castle. Computing resources for model runs in this work were provided by the Microsoft Azure high performance computers. This research under the Terrain-AI project (SFI 20/SPP/3705) has been supported by Science Foundation Ireland Strategic Partnership Programme and co funded by Microsoft.

## Author Contributions

Conceptualization, K. I. and R. F.; methodology, K.I. and R.F.; software, K.I. and R. F, with contributions by P. L. and D. W. ; validation, K. I.; formal analysis, K. I.; investigation, K. I.; resources, K.I. and R.F.; data curation, K. I.; writing—draft preparation and review, was led by K. I., G. M., M.D. and R. F., with contributions from all co-authors.; visualization, K.I.; supervision, R.F. and G. M.; project administration, R.F. and T.M.; funding acquisition, R.F. and T.M.

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

Table 1. Summary of locations of in situ measurements. The station elevation data are
obtained from Met Eireann service. The station soil texture data for Johnstown Castle and
Dripsey are obtained from previous work (Kiely et al., 2018; Murphy et al., 2022), and soil
texture map from the Irish Soil Information System (Creamer et al., 2014) are used for the in
situ Terrain-AI sites. The soil drainage classes are also obtained from the Irish soil information
database.

| Sites | Lon/Lat (°) | Elevation (m) | Field capacity | Soil texture category | | | Drainage class | Definition |
| | | | | In-situ | STATSGO | SOILGRIDS | | |
| --- | --- | --- | --- | --- | --- | --- | --- | --- |
| Athenry | -8.786/ 53.2892 | 40.0 | - | Loam | Loam | Loam | Well | Brown earth soil group, allowing water movement through the soil at a moderate rate |
| Ballyhaise | -7.309/ 54.0513 | 78.0 | - | Loam | Clay-Loam | Loam | Poor | Surface water gley soils, retaining more water at or near the surface |
| Claremorris | -8.992/ 53.7108 | 68.0 | - | Sandy-Loam | Loam | Loam | Well | Brown earth soil group, allowing water movement through the soil at a moderate rate |
| Dunsany | -6.660/ 53.5158 | 83.0 | - | Loam | Loam | Loam | Moderate | Luvisol soils, often well-drained in the upper layers |

| | | | | | | | | and slower movement deeper down. |
|---|---|---|---|---|---|---|---|---|
| Valentia | -10.244/ 51.9397 | 25.0 | - | Sandy-Loam | Sandy-Loam | Loam | Well | Brown podzolic soils, draining relatively well in the upper layers |
| Johnstown Castle | -6.505/ 52.2981 | 52.0 | 0.32 | Sandy-Loam | Loam | Loam | imperfect | Luvisol soils, often well-drained in the upper layers and slower movement deeper down. |
| Dripsey | -8.752/ 51.9867 | 190.0 | 0.42 | Loam | Loam | Loam | Poor | Surface water gley soils, retaining more water at or near the surface |

Table 2. Percentage proportion of grids covered by soil texture categories for STATSGO and SOILGRIDS databases used.

| Soil texture | STATSGO (%) | SOILGRIDS (%) |
|---|---|---|
| Sandy-Loam | 16.4 | 27.0 |
| Loam | 57.8 | 71.5 |
| Sandy Clay Loam | 0 | 1.4 |
| Clay Loam | 19.5 | 0.1 |
| Clay | 6.3 | 0 |

Table 3. Summary of NOAH-MP physical options used in this study

| Physical processes | Options |
|---|---|
| Vegetation | (4) Prescribed LAI + Prescribed max FVEG |
| Canopy stomatal resistance | (2) Jarvis |
| Soil moisture factor | (1) Noah |
| Runoff and groundwater | (3) Noah (free drainage) |
| Surface layer drag | (1) Monin-Obukhov |
| Radiation transfer | (3) Gap=1-FVEG |
| Snow surface albedo | (2) CLASS |
| Precipitation partition | (1) Jordan (1991) |
| Lower boundary soil temperature | (2) Soil temperature at 8 m depth |
| Snow/soil temperature time | (1) Semi-implicit |
| Surface resistance | (1) Sakaguchi and Zeng (2009) |

| Soil data | (1) Dominant soil texture |
| | (3) Soil composition and Pedotransfers |
| PedoTransfers | (1) Saxton and Rawls (2006) |

Table 4. Definitions of drought categories based on Relative Soil Moisture (RSM) percentiles

| ID | RSM percentile | Descriptions |
|---|---|---|
| **Dryness** | | |
| D0 | ≤ 30 | Abnormal |
| D1 | ≤ 20 | Moderate |
| D2 | ≤ 10 | Severe |
| D3 | ≤ 5 | Extreme |
| D4 | ≤ 2 | Exceptional |
| **Wetness** | | |
| W0 | ≥ 70 | Abnormal |
| W1 | ≥ 80 | Moderate |
| W2 | ≥ 90 | Severe |
| W3 | ≥ 95 | Extreme |
| W4 | ≥ 98 | Exceptional |

Table 5. Performance statistics of Relative Soil Moisture (RSM) for various soil texture categories at the topsoil (0 – 10 cm) and subsurface (0 – 100 cm) in STATSGO and SOILGRIDS for 2018 year. The errors are the median grid values. SL- Sandy Loam, L – Loam, SCL – Sandy Clay Loam, CL – Clay Loam, C – Clay.

| Soil texture | RMSD | | PBIAS | | R | |
|---|---|---|---|---|---|---|
| | STATSGO | SOILGRIDS | STATSGO | SOILGRIDS | STATSGO | SOILGRIDS |
| **Surface** | | | | | | |
| SL | 0.016 | 0.016 | -3.0 | 5.3 | 0.82 | 0.80 |
| L | 0.018 | 0.018 | -7.8 | -4.5 | 0.84 | 0.84 |
| SCL | - | 0.017 | - | -6.0 | - | 0.84 |

| Soil texture | RMSD | | PBIAS | | R | |
|---|---|---|---|---|---|---|
| | STATSGO | SOILGRIDS | STATSGO | SOILGRIDS | STATSGO | SOILGRIDS |
| CL | 0.016 | 0.016 | 11.0 | 4.6 | 0.79 | 0.86 |
| C | 0.017 | - | 9.7 | - | 0.82 | - |
| **Subsurface** | | | | | | |
| SL | 0.016 | 0.015 | 2.9 | 3.6 | 0.56 | 0.61 |
| L | 0.016 | 0.015 | -1.9 | -0.5 | 0.57 | 0.59 |
| SCL | - | 0.015 | - | 2.0 | - | 0.62 |
| CL | 0.014 | 0.015 | 4.5 | -3.3 | 0.62 | 0.58 |
| C | 0.014 | - | -1.3 | - | 0.61 | - |

Table 6. Performance statistics of Relative Soil Moisture (RSM) for various soil texture categories at the topsoil (0 – 10 cm) and subsurface (0 – 100 cm) in STATSGO and SOILGRIDS for 2019 year. The errors are the median grid values.  SL- Sandy Loam, L – Loam, SCL – Sandy Clay Loam, CL – Clay Loam, C – Clay.

| Soil texture | RMSD | | PBIAS | | R | |
|---|---|---|---|---|---|---|
| | STATSGO | SOILGRIDS | STATSGO | SOILGRIDS | STATSGO | SOILGRIDS |
| **Surface** | | | | | | |
| SL | 0.015 | 0.016 | 3.6 | 9.8 | 0.68 | 0.66 |
| L | 0.016 | 0.016 | 1.2 | 5.2 | 0.72 | 0.71 |
| SCL | - | 0.016 | - | 4.8 | - | 0.67 |
| CL | 0.019 | 0.018 | 21.2 | 18.0 | 0.61 | 0.81 |
| C | 0.019 | - | 20.1 | - | 0.79 | - |
| **Subsurface** | | | | | | |
| SL | 0.013 | 0.012 | 17.8 | 16.7 | 0.61 | 0.63 |
| L | 0.011 | 0.012 | 13.8 | 16.4 | 0.68 | 0.71 |
| SCL | - | 0.013 | - | 19.1 | - | 0.73 |
| CL | 0.013 | 0.011 | 20.5 | 16.1 | 0.73 | 0.76 |
| C | 0.012 | - | 16.1 | - | 0.77 | - |

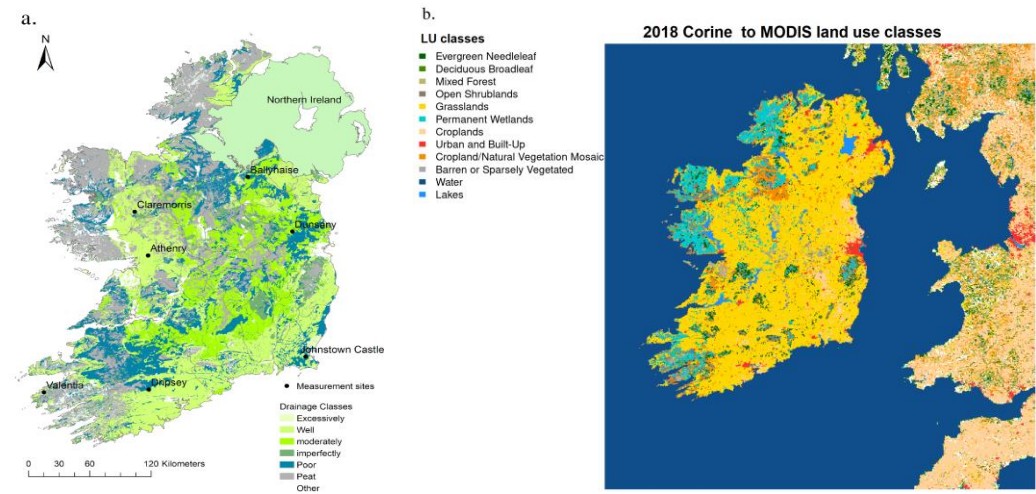

2  Figure 1. [a] Geographical locations of the selected in situ grassland sites overlaid on Ireland's
3  map of soil drainage categories. [b] Refined map of 2018 Corine to MODIS land cover classes
4  for the study domain.

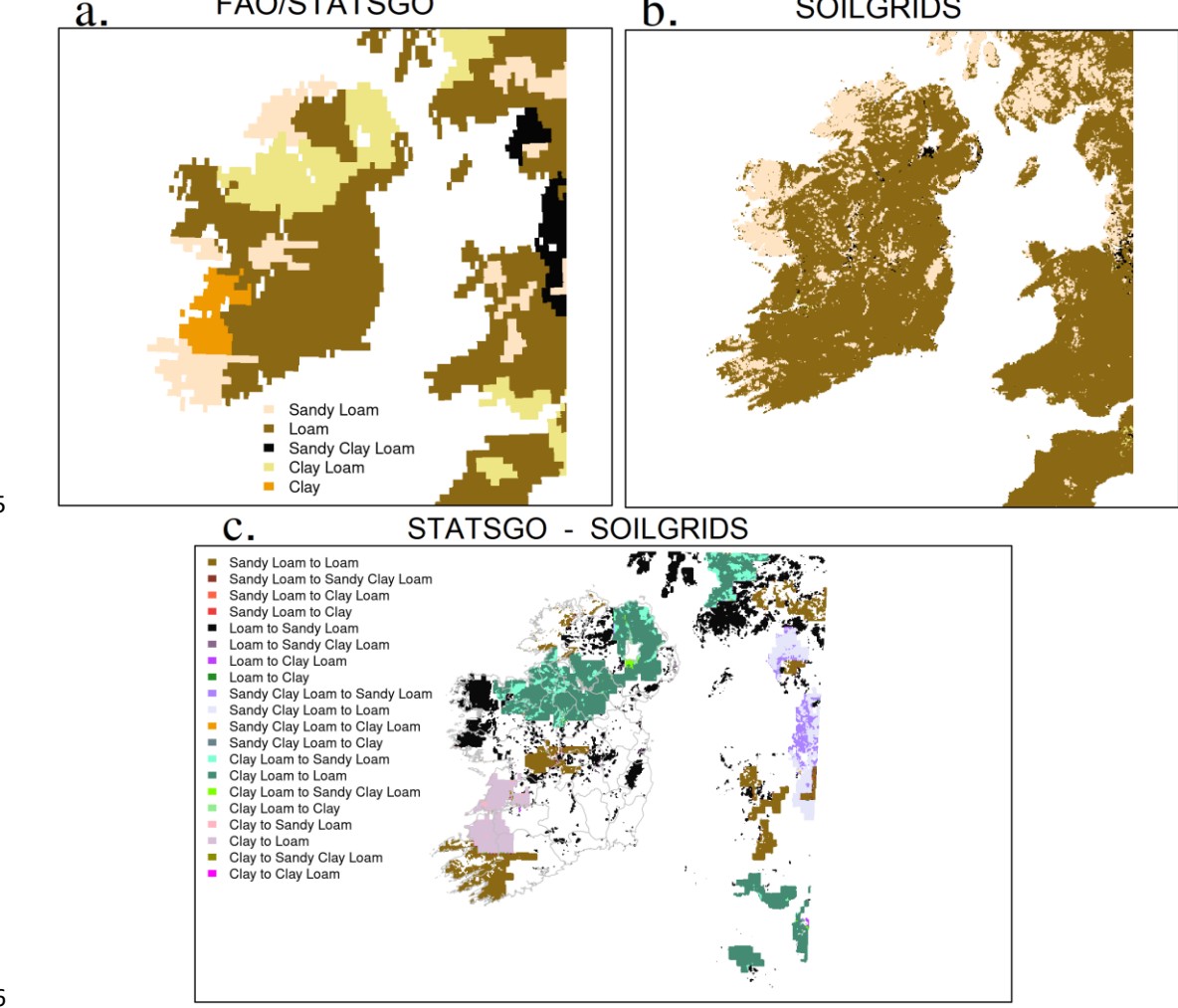

7  Figure 2. [a-b] Soil textural classes for the study domain based on global soil databases, namely
8  FAO/STATSGO and SOILGRIDS. [c] Spatial differences in the soil texture categories between
9  STATSGO and SOILGRIDS, indicating increasing or decreasing soil grain size.

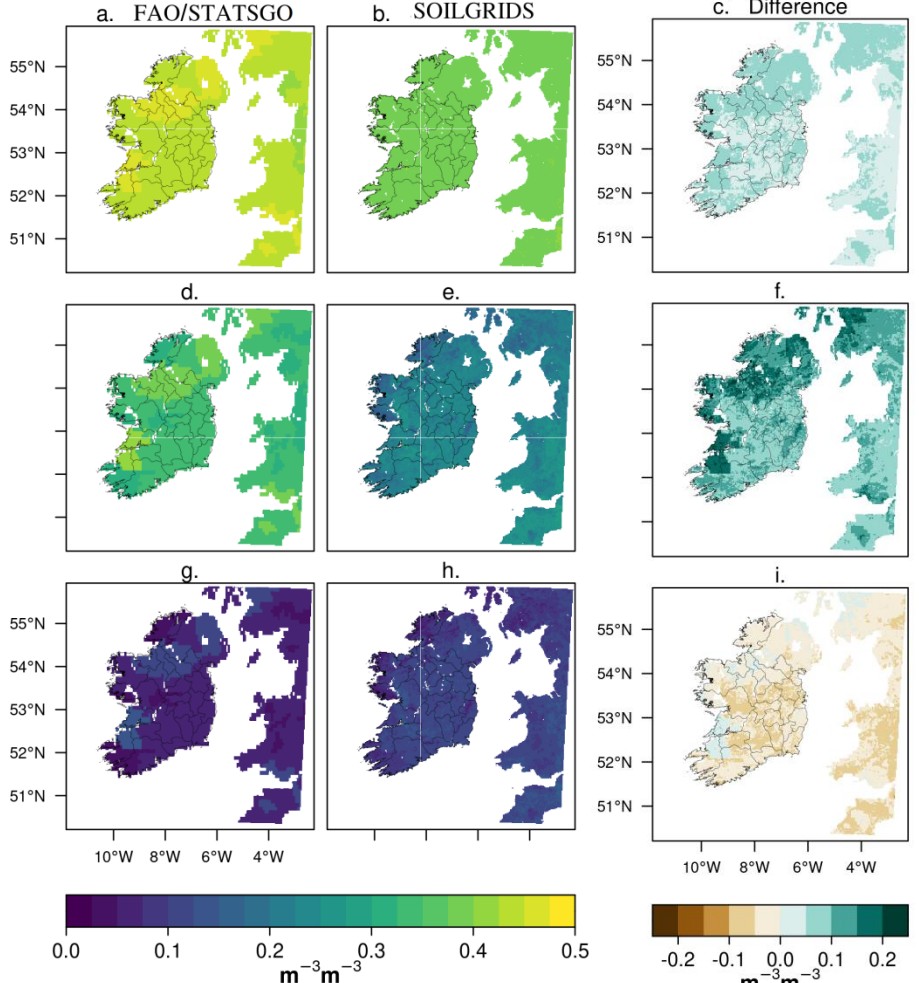

2 Figure 3. Spatial characteristics of absolute and difference between STATSGO and
3 SOILGRIDS for [a-c] soil porosity, [d-f] field capacity and  [g-i] wilting point.

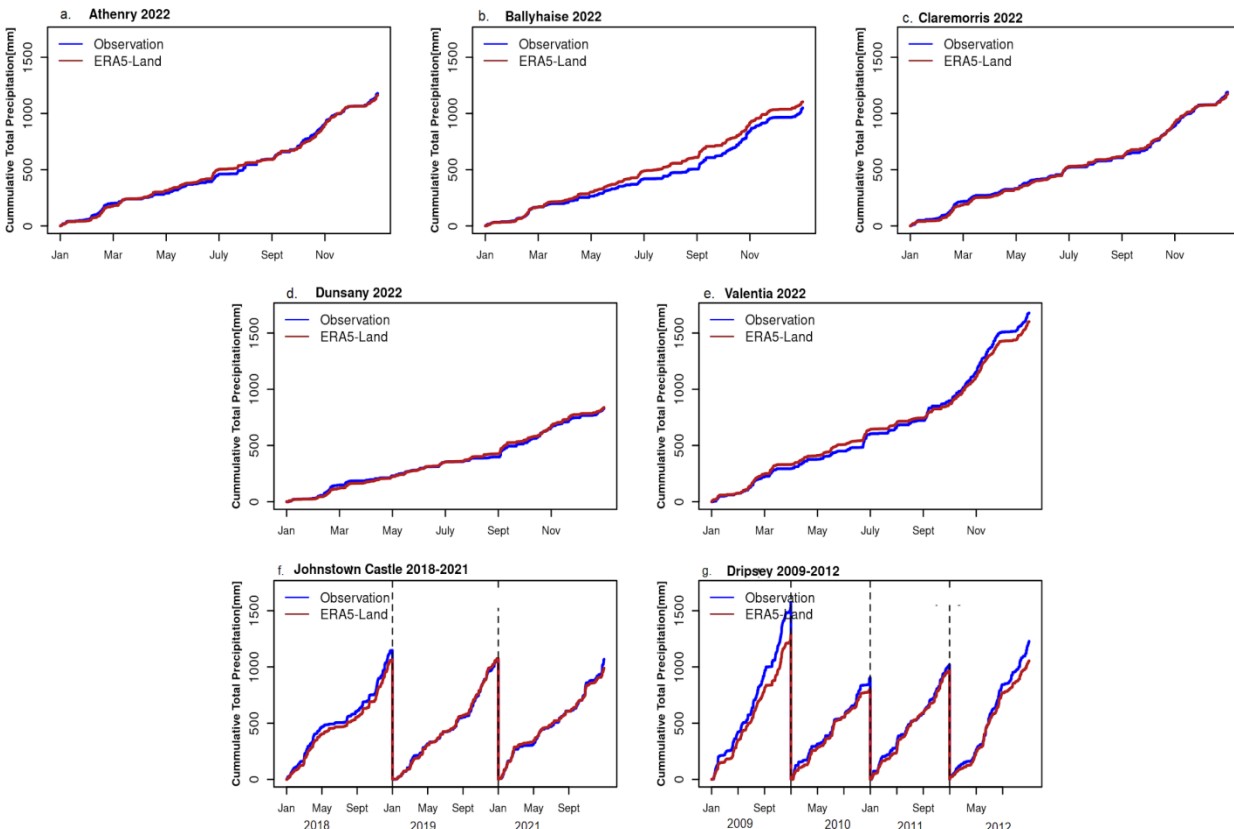

2   Figure 4. Temporal comparisons of observed total annual cumulative precipitation at the
3   selected reference stations, against the ERA5-Land colocated grids.
4

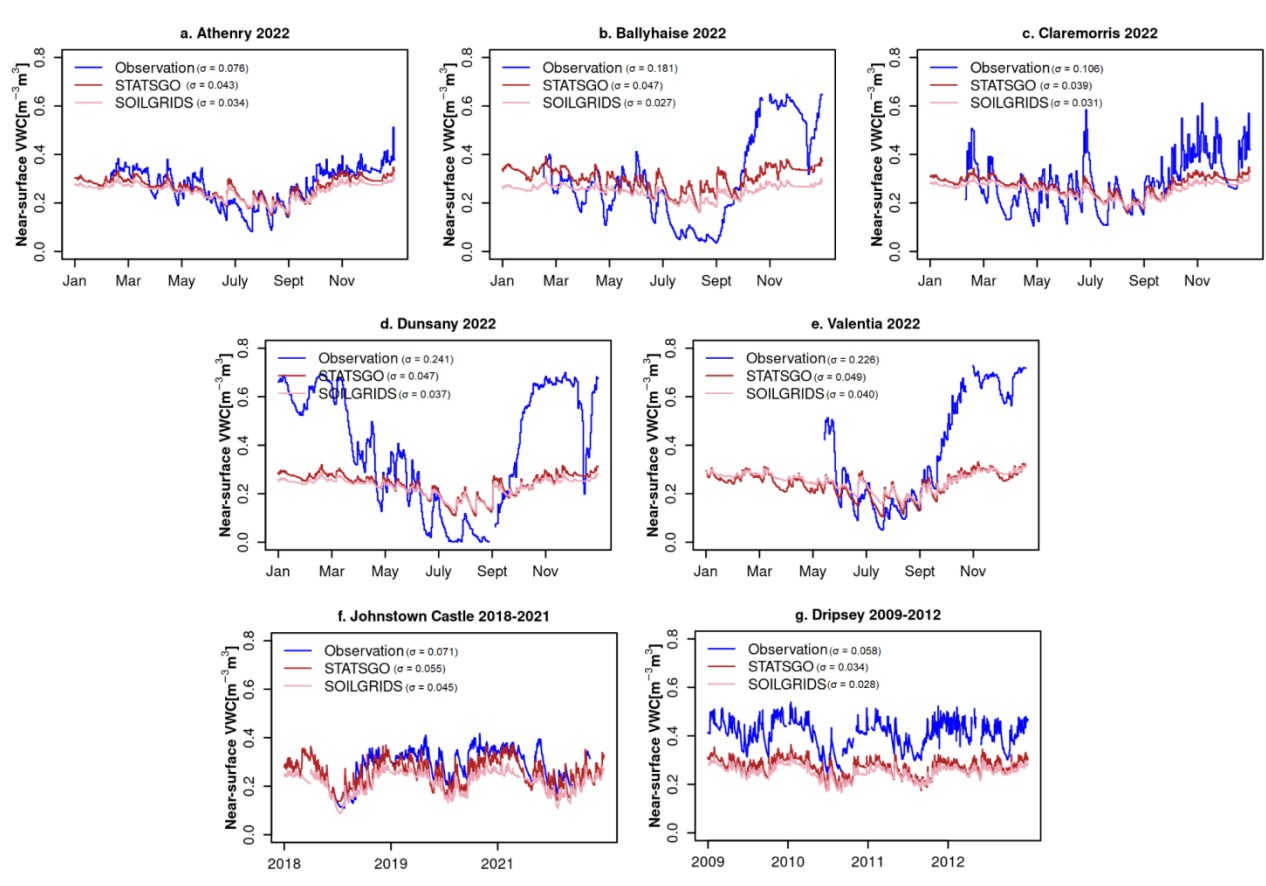

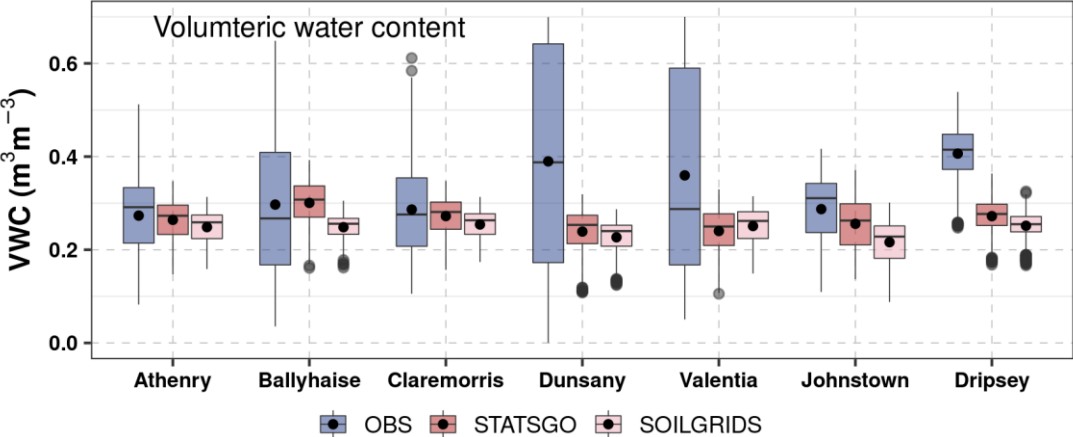

2 Figure 5. [a-g] Temporal comparisons of near-surface volumetric water contents and
3 boxplots of data distribution, between observations at 5 cm and simulated values at 0-
4 7 cm layer for the selected reference stations. For Johnstown Castle and Dripsey [f-g],
5 the model simulations are evaluated against the available observations at the top 20
6 cm depth. The black dots in the boxes represent the mean values.

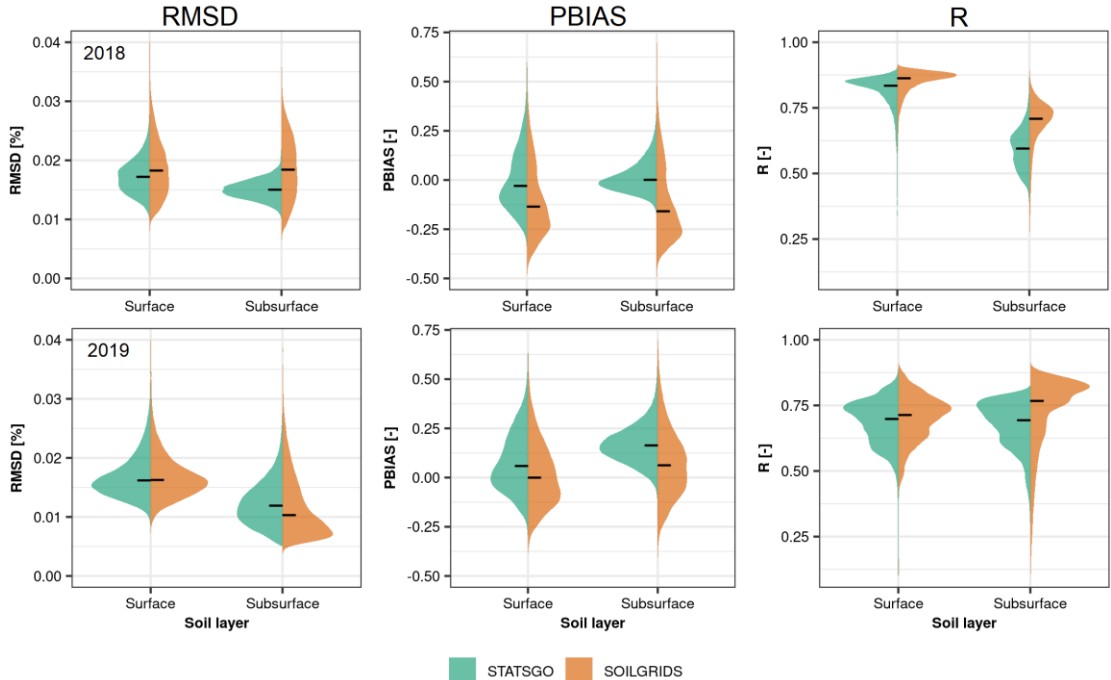

8 Figure 6. Performance statistics for STATSGO and SOILGRIDS derived Relative Soil
9 Moisture (RSM) values at the topsoil layer (0-7 cm) and subsurface soil layer (0-100
10 cm), against satellite-based ASCAT Soil Water Index (SWI), for 2018 (top) and 2019
11 (bottom) years.  N = 131,000 cells and the black crossbars are the median values.
12
13

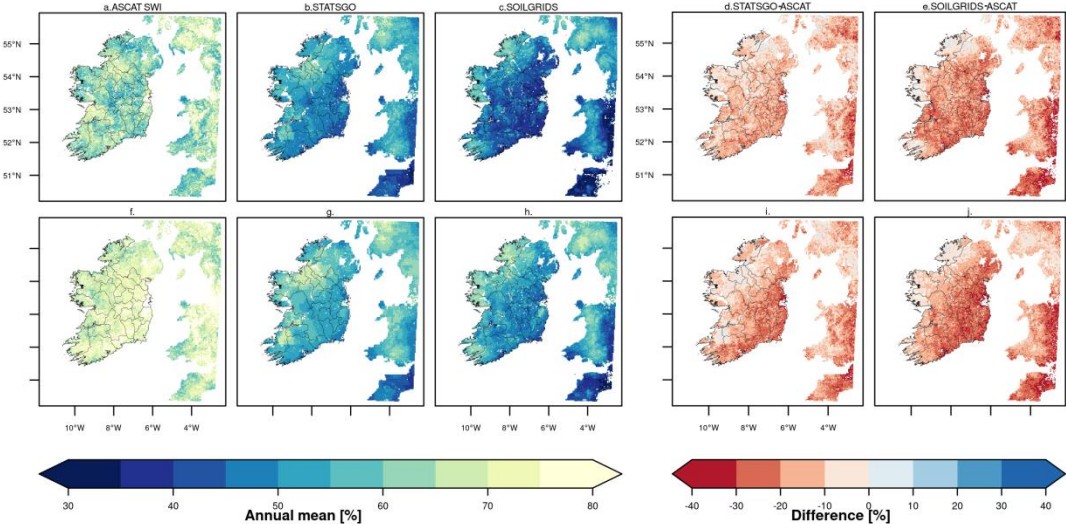

Figure 7. Spatial characteristics of absolute and difference between satellite-based
annual ASCAT Soil Water Index (SWI) and model derived annual mean Relative Soil
Moisture (RSM) at the surface , for [a-e] 2018 and [f-j] 2019 years

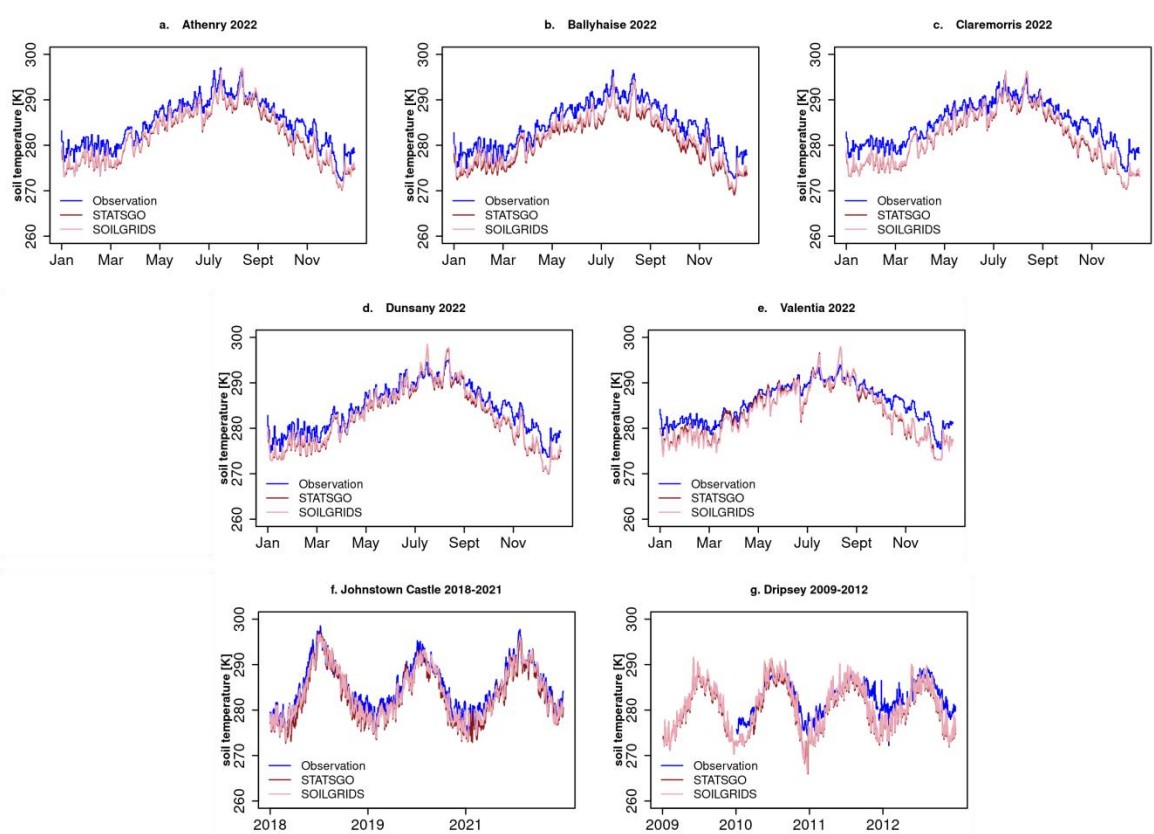

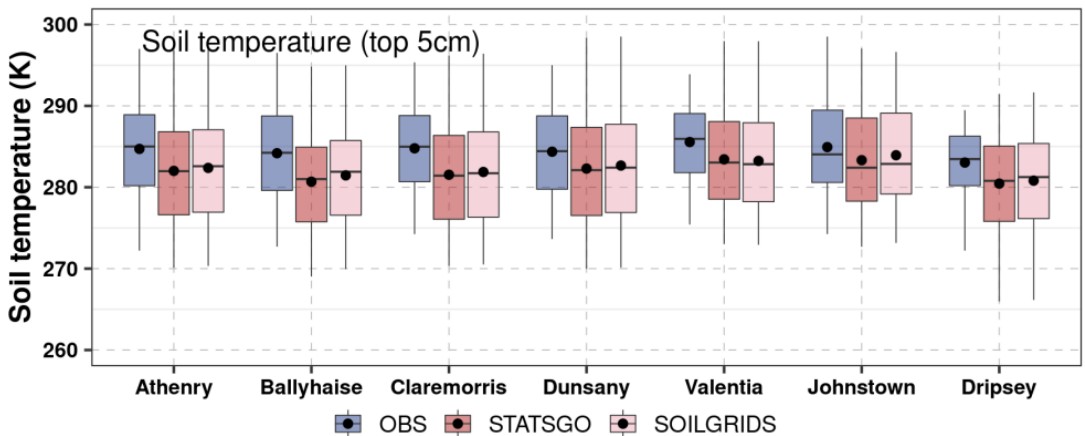

Figure 8. [a-g] Temporal comparisons of soil temperature and boxplots of data distribution, between observations and simulated values for the selected reference stations. The black dots in the boxes represent the mean values

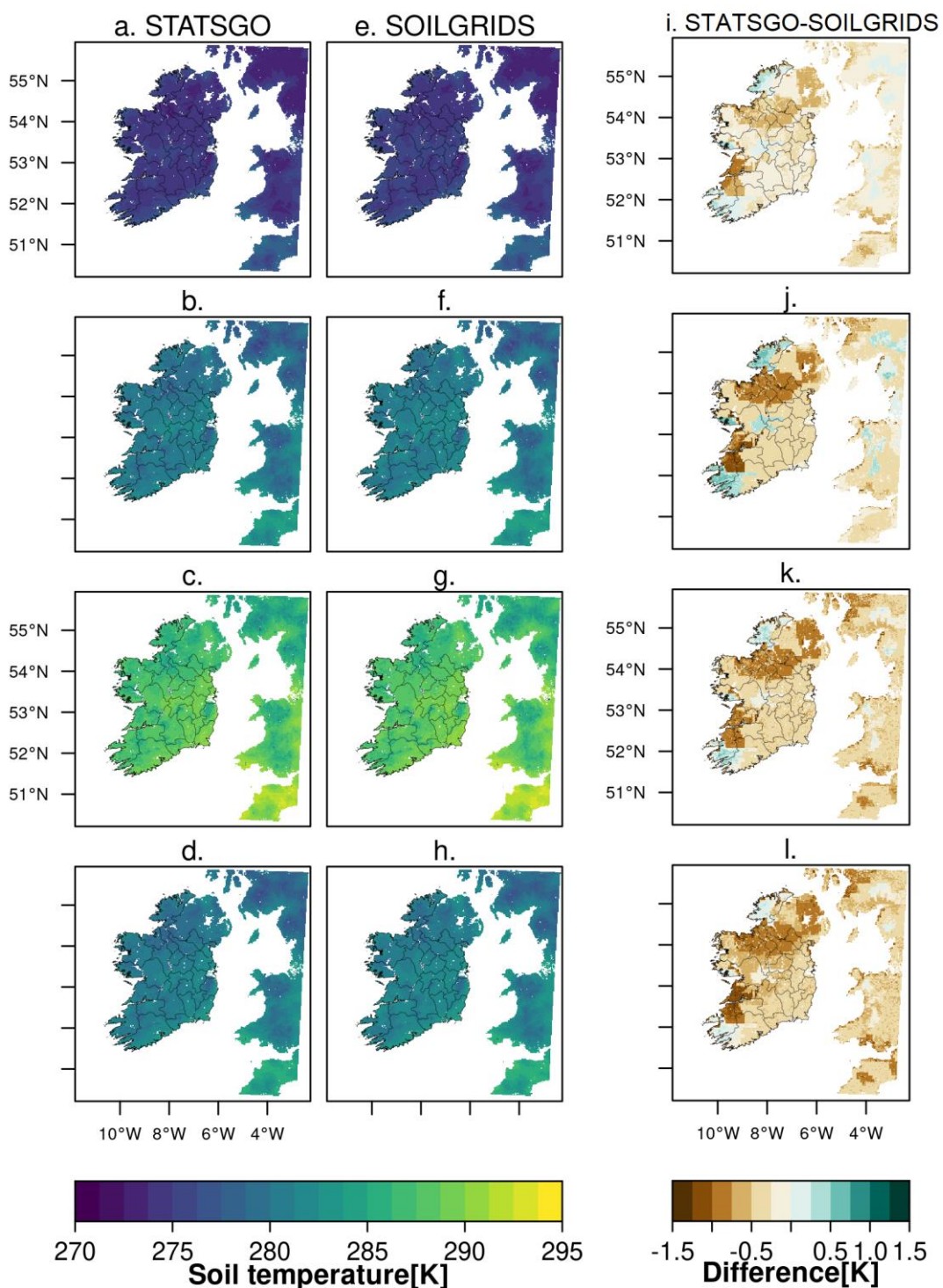

Figure 9. Spatial and seasonal characteristics of simulated top (0-7 cm) soil temperature using STATSGO [a-d], SOILGRIDS [e-h] and the difference [i-l], for the period 2009 - 2022. Rows [1-4] represent the Winter to Autumn seasons in that order.

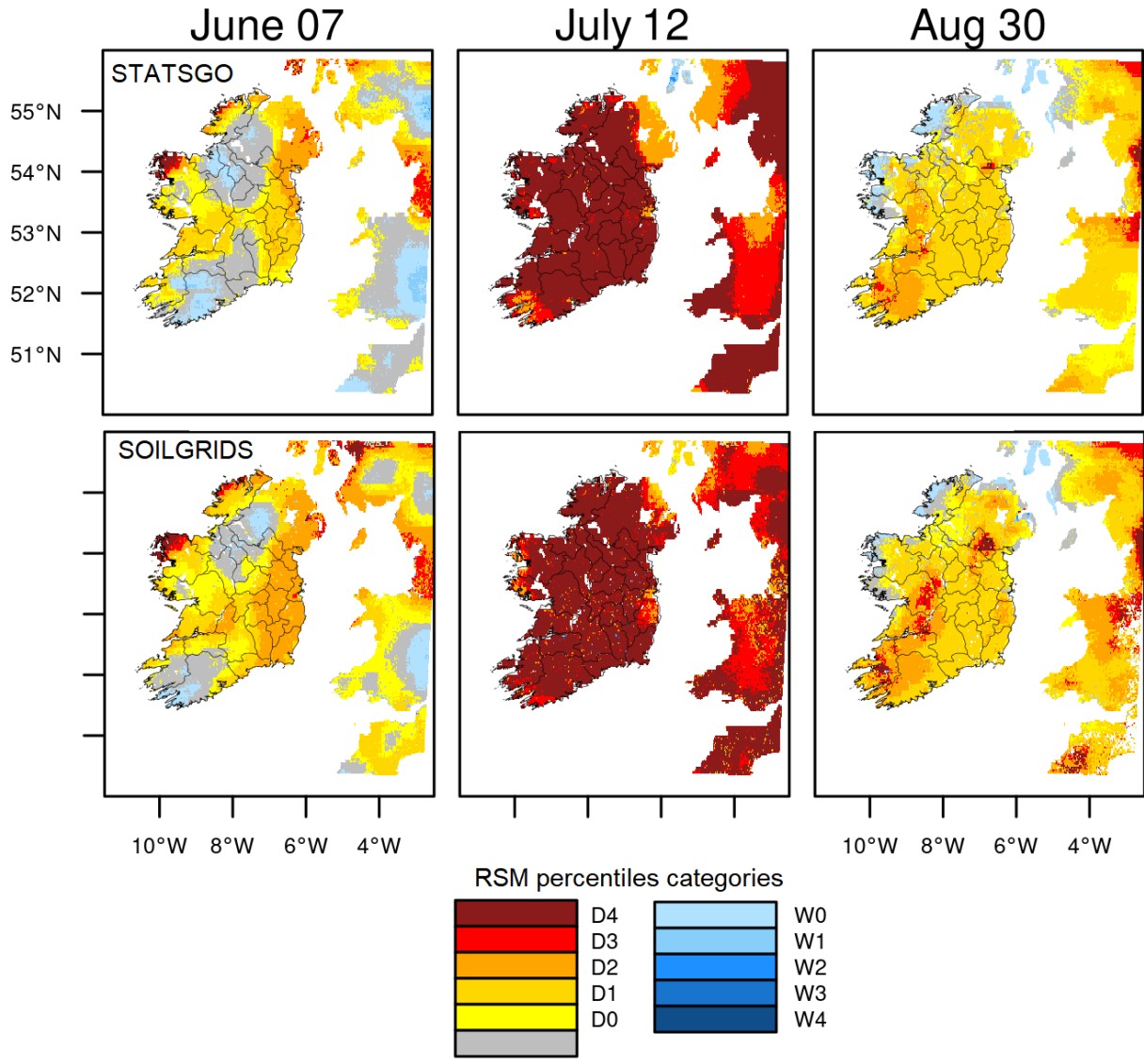

Figure 10. Spatial characteristics of soil moisture drought categories derived using 0 – 100 cm Relative Soil Moisture percentiles for STATSGO [top] and SOILGRIDS [bottom] for 2018 summer. D0-D4 represents abnormally dry, moderate, severe, extreme and exceptional droughts, while W0-W4 is the corresponding wetness categories.

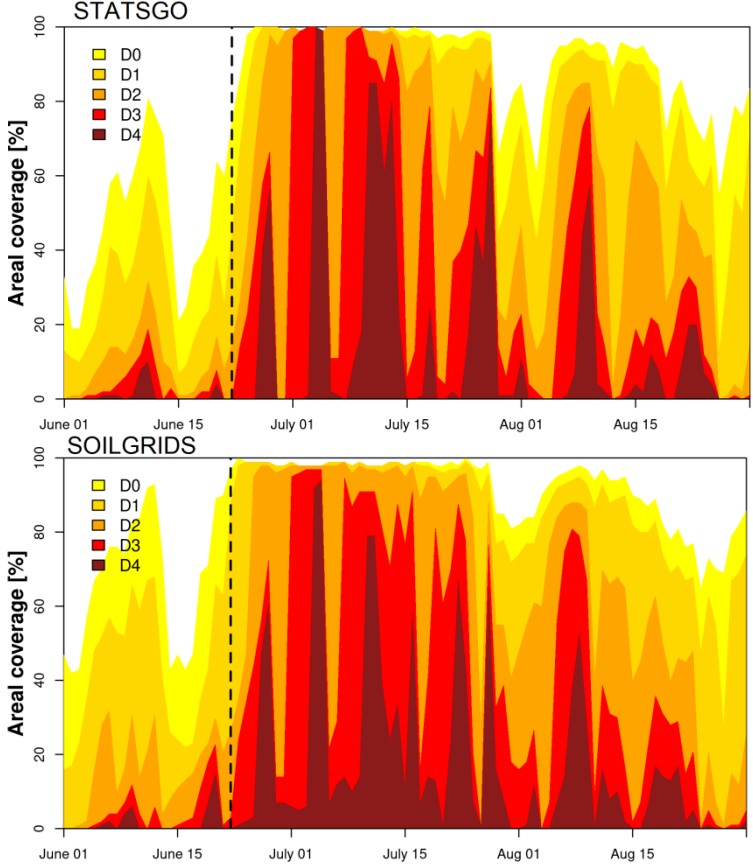

Figure 11. Time-areal coverage crossection of drought evolution based on daily 0 – 100 cm Relative Soil Moisture (RSM) percentiles during 2018 summer for STATSGO [top] and SOILGRIDS [bottom].  D0-D4 represents abnormally dry, moderate, severe, extreme and exceptional droughts. The dashed vertical lines represent the effective start of severe to exceptional droughts.

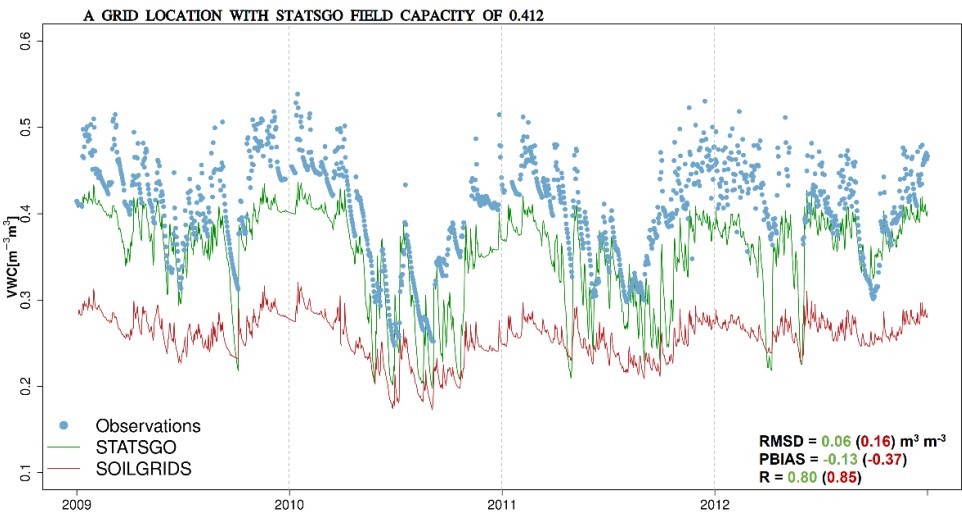

Figure 12. Temporal comparisons of observed volumetric water content (VWC) at Dripsey site, against the simulated values for a nearby grid location with field capacity of 0.412 $m^3\ m^{-3}$.

Appendix

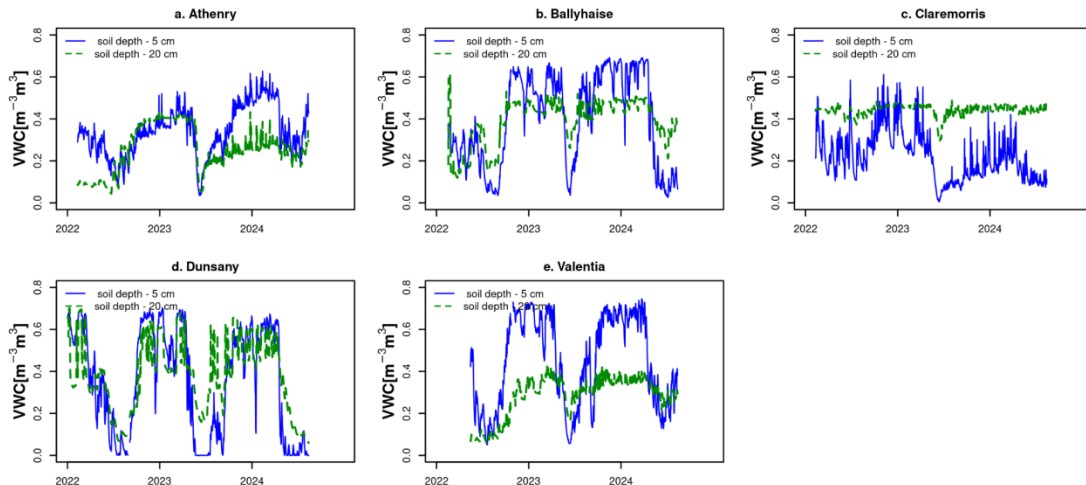

3 Figure A1. Observed 5 cm and 20 cm depths TDR soil moisture from 2022 to present
4 across the Terrain-AI stations

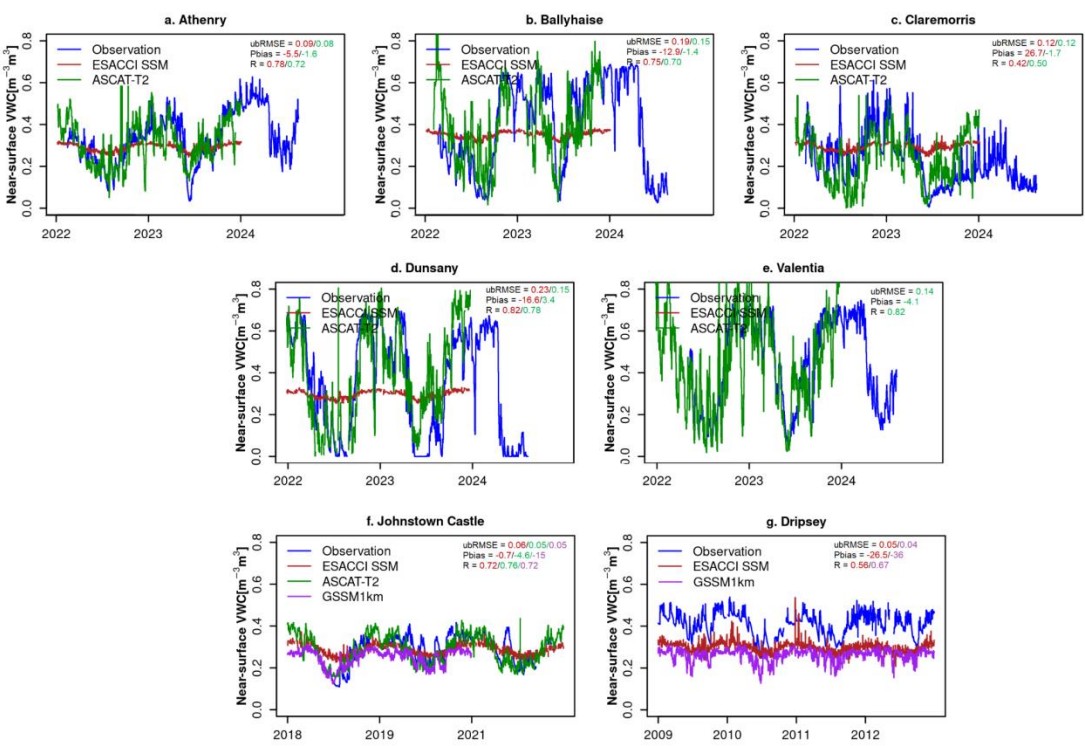

Figure A2. Evaluation of satellite-derived 1 km ASCAT-T2 (0-10 cm), 1 km GSSM (0-
5 cm) and 25 km ESACCI near-surface soil moisture against the station observations.
No available ESACCI SSM grid values for Valentia, and due to ASCAT later year of
operation in 2015, no available ASCAT values also for Dripsey.

To evaluate ASCAT SWI, we rescaled the units in percent to match the observed VWC
and other products (in $m^3 m^{-3}$) used . To achieve this, we used the variance matching
approach (equation A1) so that the linearly transformed $SWI_*$ data would have the
same mean ($\mu$) and standard deviation ($\sigma$) as the ground VWC measurements (Paulik
et al., 2014; Bauer-Marschallinger et al., 2018).

1. $$SWI_* = \frac{SWI(t) - \mu_{SWI}}{\sigma_{SWI}} \sigma_{VWC} + \mu_{VWC} \qquad\qquad (A1)$$

2.

3. As demonstrated in Figures A2-A3 for near-surface and sub-surface VWC, the ASCAT

4. $SWI_*$ generally yields better performance than ESA CCI 25 km SSM and GSSM 1 km

5. products, though the latter products show higher temporal dynamics as shown by the

6. higher temporal correlations with the ground observations. The rising and falling trends

7. are also better captured by ASCAT. Compared to ASCAT, the ESA CCI SSM and

8. GSSM show fewer fluctuations in VWC, looking very close to the subsurface VWC

9. profiles (e.g. Figure A2f). While the uncertainty in GSSM products is likely linked to

10. lack of training data from Ireland, the biases in ESA CCI SSM may be attributed to its

11. native grid resolution which is too coarse to effectively represent the soil heterogeneity,

12. and/or differences in soil depths

13.

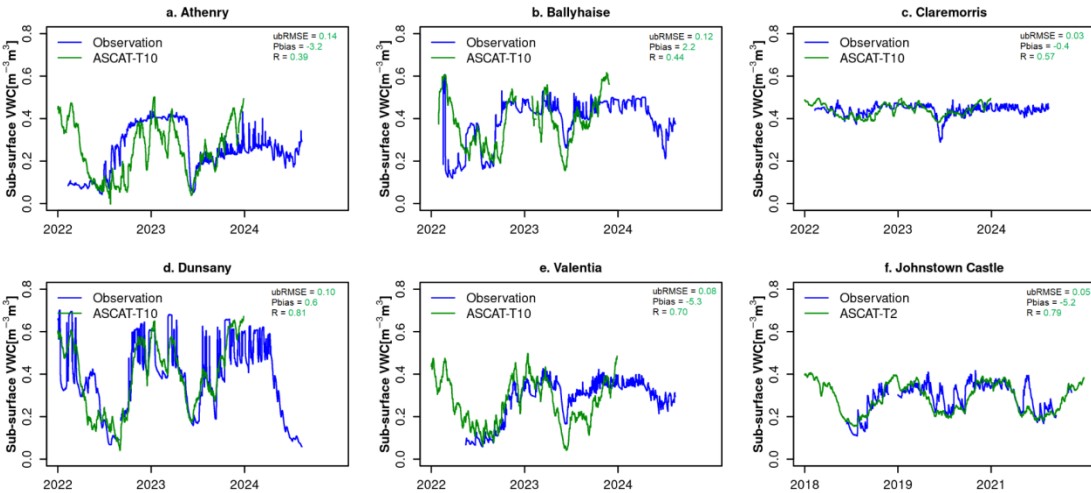

15. Figure A3. Evaluation of satellite-derived 1 km ASCAT-T10 (10-30 cm) sub-surface

16. soil moisture against the station observations (20 cm). No sub-surface values for

17. ESACCI and GSSM products

18.

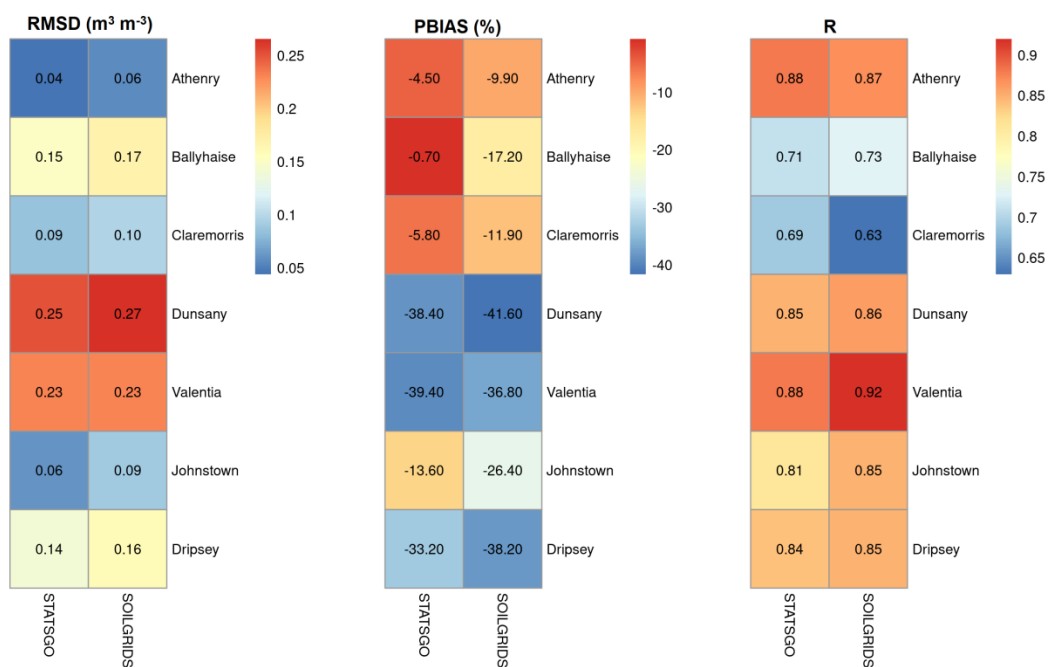

19.

Figure A4. Error statistics of volumetric water contents between observations and model experiments for the selected reference stations.

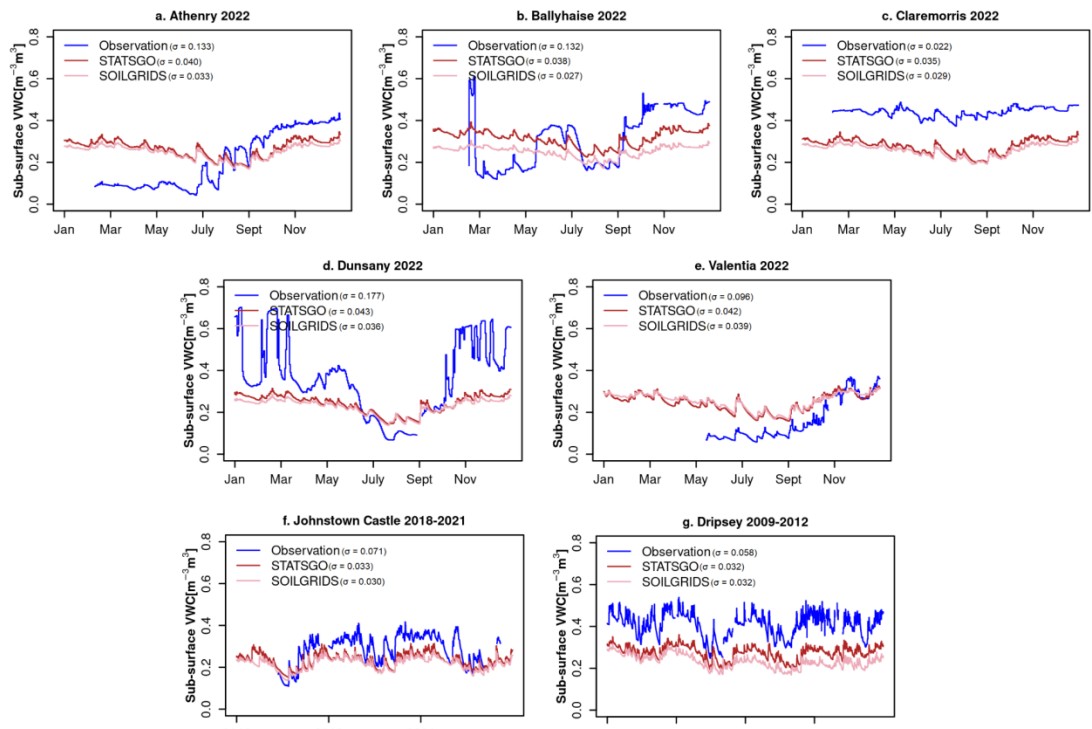

Figure A5. [a-g] Temporal comparisons of subsurface volumetric water contents between observations at 20 cm depth and simulated values at 7-21 cm layer for the selected reference stations.

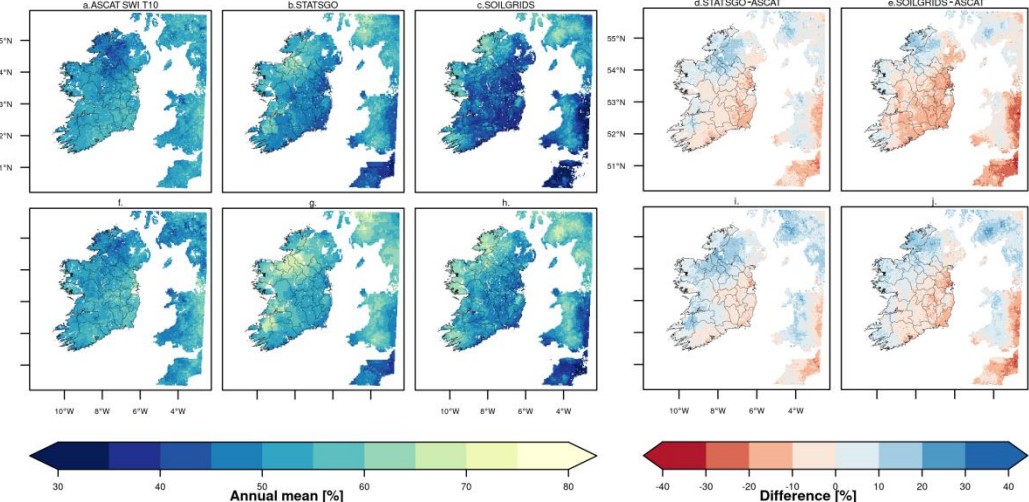

Figure A6. Spatial characteristics of absolute and difference between satellite-based annual ASCAT Soil Water Index (SWI) and model derived annual mean Relative Soil Moisture (RSM) at the subsurface , for [a-e] 2018 and [f-j] 2019 years

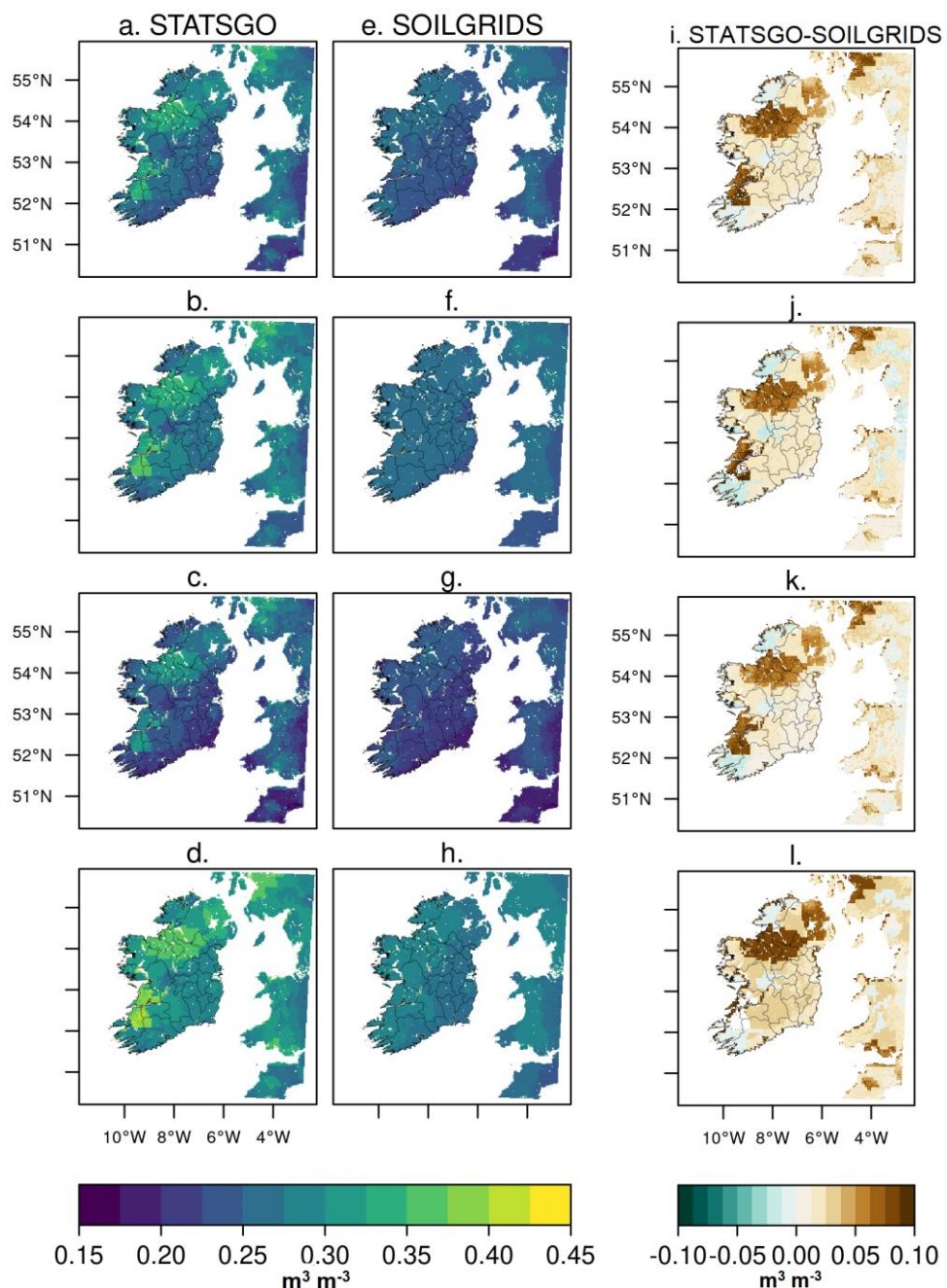

Figure A7. Spatial and seasonal characteristics of simulated top soil (0-7 cm)
volumetric water content (VWC) using STATSGO [a-d], SOILGRIDS [e-h] and the
difference [i-l], for the period 2009 - 2022. Rows [1-4] represent the Winter to Autumn
seasons in that order

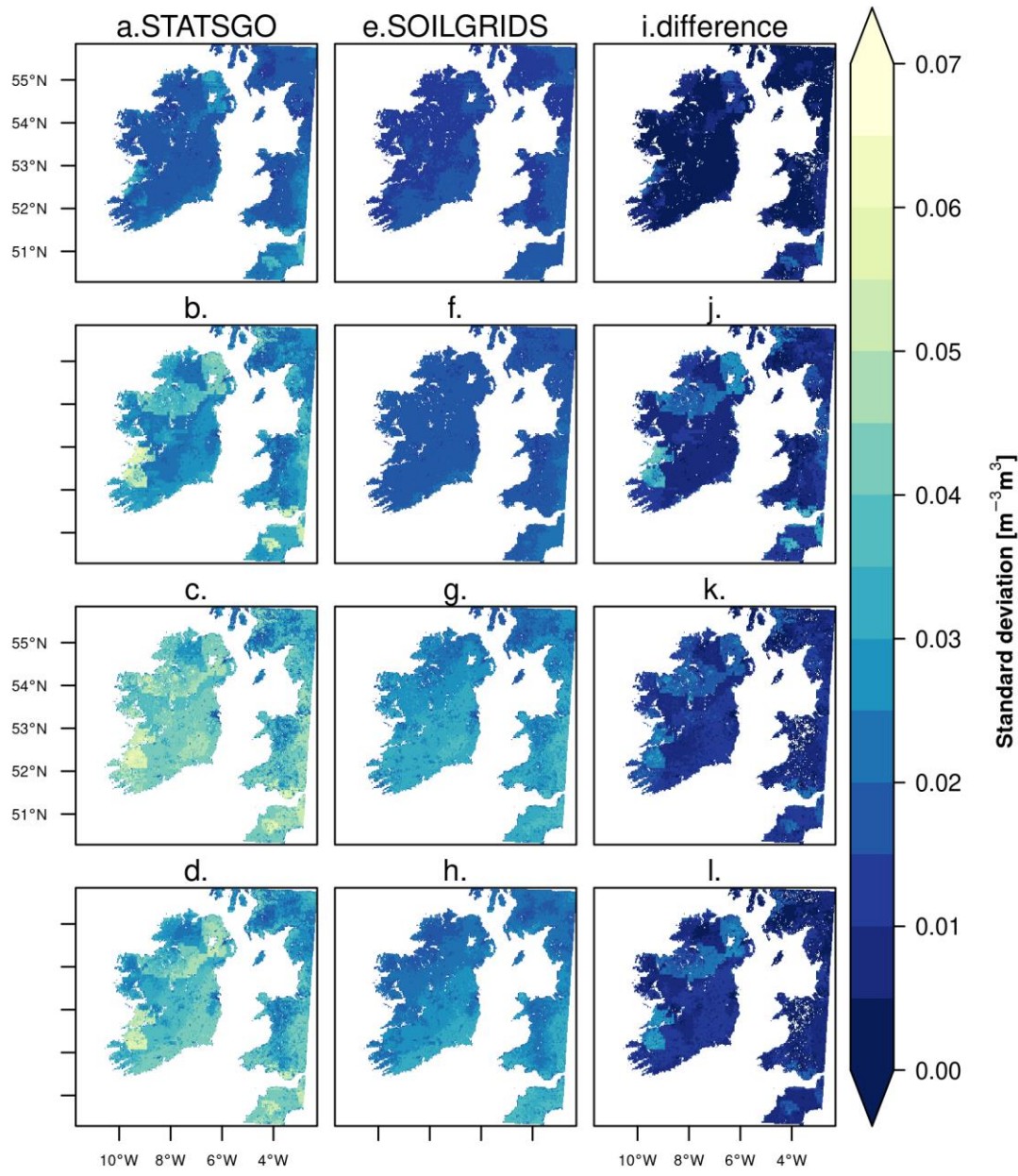

Figure A8. Spatial and seasonal characteristics of simulated long-term variability in top soil (0-7 cm) volumetric water content (VWC) using STATSGO [a-d], SOILGRIDS [e-h] and the difference [i-l], for the period 2009 - 2022. Rows [1-4] represent the Winter to Autumn seasons in that order

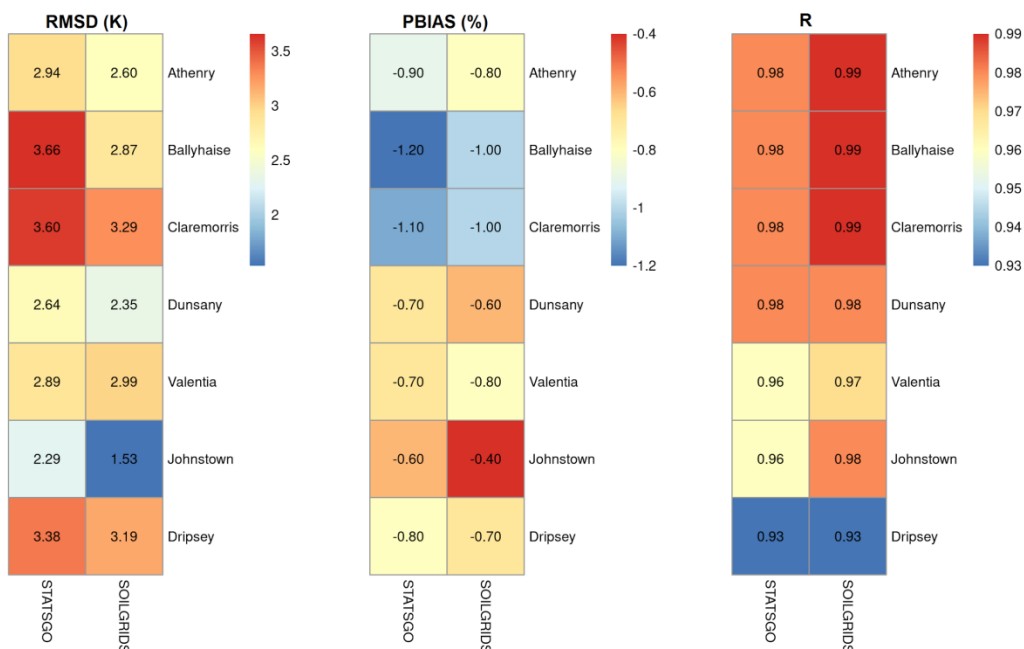

Figure A9. Error statistics of soil temperature between observations and model experiments for the selected reference stations.