# Peer review of "Implementation of global soil databases in NOAH-MP model and the effects on"

_Hydrology and Earth System Sciences, 2023_

## Referee Comment (RC2)

[referee-annotated manuscript omitted]

---

## Author Comment (AC1)

**Authors' response letter** (hess-2023-304)

The following is a point-to-point response to comments by reviewer #2

Dear Reviewer #2,

We thank you very much for your constructive comments and suggestions. The comments are very useful to improve the quality of our paper. All concerns and suggestions raised are addressed accordingly. Please find below your comments and our responses. Note that the authors' responses are highlighted in red.

P2L7

Comment #1: 'also thermal conductivity/diffusivity etc.'

Response #1: Thanks for the suggestion. This will be edited accordingly.

P2L26-28

Comment #2: 'A study on Tibetan Plateau shows that five soil databases are all different from the in-situ soil samples in terms of soil texture, bulk density, etc.'

Zhao, H., et al.: Analysis of soil hydraulic and thermal properties for land surface modeling over the Tibetan Plateau, Earth Syst. Sci. Data, 10, 1031–1061, https://doi.org/10.5194/essd-10-1031-2018, 2018

Response #2: Many thanks for sharing this useful and insightful material.

P8L1-2

Comment #3: There are many 1km soil moisture product available, perhaps interesting to add other 1km SM product to the comparison.

Han, Q., Zeng, Y., Zhang, L. et al. Global long term daily 1 km surface soil moisture dataset with physics informed machine learning. Sci Data 10, 101 (2023). https://doi.org/10.1038/s41597-023-02011-7

Zheng, C., Jia, L. & Zhao, T. A 21-year dataset (2000–2020) of gap-free global daily surface soil moisture at 1-km grid resolution. Sci Data 10, 139 (2023). https://doi.org/10.1038/s41597-023-01991-w

Zhang, Y., Liang, S., Ma, H., He, T., Wang, Q., Li, B., Xu, J., Zhang, G., Liu, X., and Xiong, C.: Generation of global 1 km daily soil moisture product from 2000 to 2020 using ensemble learning, Earth Syst. Sci. Data, 15, 2055–2079, https://doi.org/10.5194/essd-15-2055-2023, 2023

Response #3: Many thanks for sharing these important materials. We have independently evaluated and compared the 1 km GSSM soil moisture, 25 km ESACCI SSM products and 1 km ASCAT SWI against the station observations. Our results (as demonstrated in Figures 1 below) suggest that ASCAT yields better performance than ESACCI SSM and GSSM 1 km products, though the latter products show higher temporal dynamics as shown by the higher temporal correlations with the ground observations. The increasing and decreasing trends are also better captured by ASCAT. While the uncertainty in GSSM products is likely linked to lack of training data from Ireland, the biases in ESACCI SSM may be attributed to its native grid resolution which is too coarse to effectively represent the soil heterogeneity, and/or differences in soil depths. We have therefore carefully chosen ASCAT as the reference.

[Figure]

Figure 1. Evaluation of satellite-derived 1 km ASCAT-T2 (0-10 cm), 1 km GSSM (0-5 cm) and 25 km ESACCI near-surface soil moisture against the station observations. No available ESACCI SSM grid values for Valentia, and due to ASCAT later year of operation in 2015, no available ASCAT values also for Dripsey.

P9L30

Comment #4: There are descriptions on which soil datasets lead to better statistics, however, this reviewer is missing 'then what'. What readers can learn from this exercise in terms of improving either one soil databases? Or indicating model deficiency for future improvements?

Response #4: Thanks for the comment. In our conclusions, we highlight the shortcomings of global soil databases, and model deficiency due to uncertainty of soil hydro-physical parameters. Our study underscores the need to improve the PTFs used in Land surface models and/or development of detailed region-specific soil properties to enhance land surface model performance applied to highly heterogeneous and managed landscapes and improve our understanding of the role of soils in land surface processes.

P9L35

Comment #5: It is not clear what is the resolution of the model areal grid. Is it 9km (5 arcmin)?

Response #5: The model resolution is 1 km. This is highlighted in section 2.4 of the paper.

P10L22

Comment #6: For Athenry and Johnstown Castle, you can say that, for the rest, the simulations are way off the observations.

Response #6: We agree with your observations. We have mentioned these locations in the same statement, but we may rephrase the statement accordingly to avoid confusion.

P10L30-31

Comment #7: By 'reference' here, do you mean, in this case, due to the match of grid resolution, ASCAT satellite data is more comparable to the model simulation results?

Response #7: Yes, ASCAT is treated as the reference, as it performs better than ESA CCI SSM and 1km GSSM products (see Figure 1 above).  It's also non-blended/non-modeled satellite products, 1 km spatial resolution matching the model grid and provides information that allows evaluation of surface and subsurface soil moisture.

P12L17-19

Comment #8: This reviewer think it is still making sense to compare with satellite LST data, for example, from ESA CCI LST. Such comparison with independent dataset may reveal some further insights.

Response #8: Thanks for your useful suggestion. We are aware of this and have previously incorporated satellite LST from MODIS to investigate the capability of a land surface model over Ireland (Ishola et al. 2022). The satellite LST allowed us to compare and demonstrate the patterns of change in temperature across the surface, however, the context of the current work is on the physics of soil moisture and soil temperature for which we now have actual measurements from a network of stations to evaluate our models across the country. We believe this is a fair judgement and therefore we have ruled out further investigations using satellite LST

Reference

Ishola, K. A., Mills, G., Fealy, R. M., Fealy, R.: A model framework to investigate the role of anomalous land surface processes in the amplification of summer drought across Ireland during 2018, I*nternational Journal of Climatology*, 43, 480 - 498. https://doi.org/10.1002/joc.7785 , 2022

P14L14-17

Comment #9: From the result, it is not clear how field capacity difference influences the differences in soil moisture simulations.

Response #9: Thanks for the comment. From basic knowledge of soil water, we understand that field capacity and wilting point are required to determine the available water, the model implements that within the diffusivity form of Richard's soil water representation shown below and in this work we have provided a demonstration of the importance of the soil datasets field capacity in driving the soil water conditions within the model.

As demonstrated at Dripsey site (see Figure 12 in the paper) for example, when we changed the field capacity to be representative of the actual value of the site, we obtained much better results for STATSGO, relative to SOILGRID. The SOILGRID with lower soil field capacity than STATSGO (see Figure 3 of the paper) allows water to drain more quickly, thereby retaining less water in the pore spaces and increasing dry conditions in the model. Our calculation of grid-scale Pearson correlation coefficients between mean difference (standard deviation difference) of soil moisture and difference in the field capacity yields approximately 0.65 (0.45), and -0.40 (-0.1) with the soil hydraulic conductivity difference. This suggests that the mean and variability of soil moisture difference between STATSGO and SOILGRIDS are significantly ($p < 0.05$) influenced by the soil parameter values. However, for robust understanding on whether the remaining errors in the simulations are still linked to soil parameter uncertainty, it would involve performing ensemble simulations based on ensemble of parameter values which is beyond the scope of this work.

$$\frac{\partial \theta}{\partial t} = \frac{\partial}{\partial z}\left(D\frac{\partial \theta}{\partial z}\right) + \frac{\partial K}{\partial z} + S$$

P14L25-27

Comment #10: it is not clear from the result that there is a reference soil databases. So when the authors mentioned 'accurately represented', this reviewer is confused.

Response #10: The stations (Johnstown Castle and Dripsey) with extended period of measured data provide further information on soil properties (e.g. Kiely et al., 2018; Ishola et al., 2020), including the soil FC values which were contrasted against the model prescribed FC values before arriving at the phrase 'accurately represented'.

We will revise section 2.5 to include information on the available measured soil FC values from the two stations.

References

Kiely, G., Leahy, P., Lewis, C., Sottocornola, M., Laine, A., Koehler, A.-K.: GHG Fluxes from Terrestrial Ecosystems in Ireland. Research report No. 227.EPA Research Programme, Wexford. Available online at https://www.epa.ie/publications/research/climate-change/research-227.php, 2018

Ishola, K.A., Mills, G., Fealy, R.M., Ní, C.Ó. and Fealy, R.: Improving a land surface scheme for estimating sensible and latent heat fluxes above grassland with contrasting soil moisture zones. *Agricultural and Forest Meteorology*, 294,108151. https://doi.org/10.1016/j.agrformet.2020.108151 , 2020

P15L8-9

Comment #11: but this is also very much induced by how you preprocess SOILGRID data to your model grid?

Response #11: We agree with your observations. The lack of improvement in SOILGRID for this illustration is also linked to the preprocessing, particularly the empirical PTFs coefficients which are underpredicting the soil parameter values at the topsoil layer for Ireland's landscapes.

The statement will be revised accordingly.

P15L32-34

Comment #12: A latest paper explains some further challenges in terms of PTFs.

Weber, T. K. D., Weihermüller, L., Nemes, A., Bechtold, M., Degré, A., Diamantopoulos, E., Fatichi, S., Filipović, V., Gupta, S., Hohenbrink, T. L., Hirmas, D. R., Jackisch, C., de Jong van Lier, Q., Koestel, J., Lehmann, P., Marthews, T. R., Minasny, B., Pagel, H., van der Ploeg, M., Svane, S. F., Szabó, B., Vereecken, H., Verhoef, A., Young, M., Zeng, Y., Zhang, Y., and Bonetti, S.: Hydro-pedotransfer functions: A roadmap for future development, EGUsphere [preprint], https://doi.org/10.5194/egusphere-2023-1860, 2023.

Response #12: Many thanks for sharing this material with us. It's very relevant and information will be integrated into the statement made in the paper.

P17L11

Comment #13: you cannot tell from the results.

Response #13: We have made the judgement based on Athenry, Johnstown Castle and Dripsey stations, where the variances in simulated soil moisture are close to the observed values (see Figure 2 below). However, we will revise this section accordingly to properly describe the uncertainties in our results.

[Figure]

Figure 2. Temporal comparisons of near-surface volumetric water contents between observations at 5 cm depth and model simulations centered at 3.5 cm depth. The model simulations are contrasted with 20 cm depth due to unavailable near-surface observations for Johnstown Castle and Dripsey

P17L21-23

Comment #14: are you suggesting using ASCAT SWI is already enough, and there is no need of a LSM here then?

Response #14: We are suggesting based on the better performance of ASCAT SWI against station observations, relative to GSSM and ESACCI SSM (see Figure 1 above). The LSM is important to better understand the physics and complexity of soil moisture dynamics especially for Ireland where the landscapes are heavily managed and heterogeneous.

---

## Author Comment (AC2)

**Authors' response letter** (hess-2023-304)

The following is a point-to-point response to comments by reviewer #1

This work compared the simulation performance of Noah-MP land surface model over Ireland using two global soil property datasets, including a high-resolution SOILGRID data and a coarse-resolution STASGO data. The results showed that, the coarse STATSGO performs as good as the fine-scale SOILGRIDS soil database, although they both have dry biases. Overall, I think comparing the added value of high-resolution soil dataset to land surface modeling is an important and meaningful topic for both data and model developers. However, there are lots of uncertainties caused by other processes, such as the observation and the model physical parameterization, instead of the soil database. Following major issues should be carefully considered before the further consideration of the publication.

Dear Reviewer #1,

We thank you very much for your constructive comments and suggestions. The comments provided are very useful to improve the quality of our paper. All concerns and suggestions raised are addressed accordingly. Please find below your comments and our responses. Note that the authors' responses are highlighted in red.

Comment #1:

The innovation. The current innovation is somewhat weak for the HESS journal. The work is very similar to Zhang et al. (2023), although the authors said that they used the SOILGRID dataset. The work compared the difference of soil moisture/temperature simulation and drought processes (2 drought events) over Ireland, and made some conclusions. This makes the work like a technical comparison without in depth analysis of the uncertainties and related reasons. For example, why the high-resolution soil database performs similarly with the coarse-resolution? Is it because the uncertainties from soil database itself or the model structure and physical parameteritzations (for example, the uncertainties of PFT function to accurately derive the soil hydrological properties based on soil texture data)?

Response #1:

Zhang et al. (2023) did an excellent piece of work, incorporating global soil datasets into the WRF-Hydro model over southern Africa. However, our application differs in terms of climatic region, nature of the managed and highly heterogeneous soil landscapes, focus on specific weather events and soil physics. Ireland lies in the maritime temperature region with cool temperatures year round and no marked seasonality to precipitation. As a consequence, growing conditions are near optimal and grassland land cover accounts for almost 60% of the total land area. Grasslands here are also highly managed, for grazing and the provision of overwintering grass fodder for animals. Ireland also has far younger soils that are heterogeneous over small spatial scales; compared to South Africa, where soils are older and more consistent over large spatial areas.

In spite of its maritime climate, Ireland can and does experience periodic/seasonal soil moisture deficits, particularly in the sandy soils located in the south-east of the island. To the north and west, soils tend to have high clay content, which can act as a buffer to prolonged dry periods. Improved understanding of the potential impacts of climate change, specifically changes in the frequency and magnitude of drought/heat waves is of particular importance, not just for agricultural productivity, but on grasslands across Europe and more generally – grassland land cover represent an estimated 20-40% of global land cover (estimates vary depending on how grasslands are categorized).

Critically, the use of direct ground observations from sites with different soil characteristics allows us to more robustly evaluate the efficacy of the selected global soil databases in representing complex soil regimes. Compared to the work of Zhang et al., we evaluated different soil physics, including those based on PTFs, to provide insights into advancing soil hydrothermal extremes by evaluating the added benefit of vertical soil properties derived from 250 m SOILGRID maps. We also focus on the ability of the land surface model – NOAH-MP, which provides the only physical boundary to WRF climate model, to estimate soil hydro thermal properties under both mean and extreme conditions.

On basis of the background dynamic climate, grassland land cover/land cover use and soil conditions found here, and the evaluation of different soil physics schemes against a network of in-situ sensors (rather than remote), we believe the study, while strongly complementary to Zhang et al. (2023), is novel in application and relevant for a global audience.

The question about why both soil databases perform similarly was addressed in the paper (P15L21-34 and P14L18-30). We demonstrate that on one hand is a potential issue with misrepresentation of soil textural classes, but this does not fully explain why as sites with different soil texture types between STATSGO and SOILGRIDS (Table 1) (e.g. Valentia and Ballyhaise) produce similar results (Figures 5b, e in the paper). The reason for the similar performance between the soil databases is linked to the uncertainties of the empirical PTFs in the NOAH-MP model as discussed in P15L21-34. We have now carried out an uncertainty analysis as shown below (Figure 1) to further support our points.

The grid-scale uncertainty is quantified using the standard deviation difference between the experiments at the model topmost soil layer. Though model internal variability shows higher tendency with STATSGO than the SOILGRIDS, especially from Spring to Autumn, the standard deviation is generally below 0.08 $m^3m^{-3}$ and the difference between the experiments is relatively small. The spatial patterns of standard deviation difference and the mean difference (see Figure A2 in the paper) are consistent with the topsoil textural classes (see Figure 2c in the paper) and field capacity (see Figure 3f in the paper). Our calculation of Pearson correlation coefficients between mean difference (standard deviation difference) of soil moisture and difference in the field capacity yields approximately 0.65 (0.45), and -0.40 (-0.1) with the soil hydraulic conductivity difference. This suggests that the mean and variability of soil moisture difference between STATSGO and SOILGRIDS are significantly ($p < 0.05$) related to the soil parameter values. This is further confirmed at a point location (see Figure 12 in the paper). However, for a more robust understanding on whether the remaining errors in the simulations are still linked to soil

parameter uncertainty, it would involve performing ensemble simulations based on ensemble of parameter values which is beyond the scope of this work.

We will revise the paper accordingly to incorporate some of the information provided here.

[Figure]

Figure 1. Spatial and seasonal comparisons of top-soil soil moisture internal variability between STATSGO [a-d] and SOILGRIDS [e-h] for the period 2009-2022. The rows represent Winter to Autumn.

Comment #2:

The station observation. There are 6 observation stations used in this work, and 4 of them are from a new network (Terrain-AI) using Time Domain Reflectometry (TDR) sensors. I found noteworthy difference between the Terrain-AI based observations and the 2 long-time eddy covariance grass flux sites. For example, most of the Terrain-AI based observations show high values during wet seasons (larger than 0.5 m3/m3), while observations from 2 eddy covariance grass flux sites are generally lower than 0.5. I wonder whether these high-values in Terrain-AI stations are true or not? This is very important to the conclusion, as the dry bias is mainly due to these stations. If the observation is true, then why there are

such large differences considering they are all loam or loam-sand stations?

Response #2:

We agree that the Terrain-AI based soil moisture data should not be far from their counterpart flux sites, given that the soil textures, and instrumentation are similar.

At Terrain-AI, we have ensured standard, globally acceptable and well calibrated TDR sensors (Campbell Scientific CS615/CS616) across the stations. Therefore, to investigate these concerns, we analysed additional measured soil moisture records from 2023 to present across the Terrain-AI stations (Figure 1 below). In fact, the 2022 high values during the wet period are almost the same as 2023 and 2024. Hence, we should be safe to respond that the values are true and there is no evidence of sensor decay in these measurements.

However, we note that the soil moisture values at flux sites were measured in top 20 cm (Kiely et al., 2018; Murphy et al., 2022), whereas the Terrain-AI soil moisture used were measured at 5 cm depth (Figure 2 below, blue lines). Kiely et al. (2018) also mentioned that the porosity for top 5 cm at Dripsey site is 65 % which is broadly within the range of 5 cm values measured across Terrain-AI stations (Figure 2 below, blue lines). Unfortunately, we do not have near-surface soil moisture measurements from the flux sites, but we analysed the measured 20 cm soil moisture values from Terrain-AI stations (Figure 2 below, green lines), the values (0.4 - 0.45 $m^3m^{-3}$) at this depth during the wet period are broadly close to that of the flux sites. Therefore, we can state with some degree of confidence that the large differences in the wet soil moisture values between the Terrain-AI and the flux stations are due to different soil depths, as the near surface soil will be wetter than the deep soil layer between rainfall events. Again, this points to the more complicated soil landscapes experienced here, compared to Zhang et al. (2023).

For clarity, we have now separated the model evaluation analysis between near-surface (5 cm soil depth) and subsurface (top 20 cm) as shown in Figures 3 and 4 below.

We will revise the paper to provide more information on the station observations and associated soil depths. We will also incorporate the analysis in Figures 3 and 4 below to ensure clarity in relation to model evaluation and performance at different depths.

References

Kiely, G., Leahy, P., Lewis, C., Sottocornola, M., Laine, A., Koehler, A.-K.: GHG Fluxes from Terrestrial Ecosystems in Ireland. Research report No. 227.EPA Research Programme, Wexford. Available online at https://www.epa.ie/publications/research/climate-change/research-227.php, 2018

Murphy, R. M., Saunders, M., Richards, K. G., Krol, D. J., Gebremichael, A. W., Rambaud, J., Cowan, N., Lanigan, G. J.: Nitrous oxide emission factors from an intensively grazed temperate grassland: A

comparison of cumulative emissions determined by eddy covariance and static chamber methods, Agric. Ecosys. Environ., 324 107725, https://doi.org/10.1016/j.agee.2021.107725 , 2022

[Figure]

Figure 2 . Observed 5 cm and 20 cm depths TDR soil moisture from 2022 to present across the Terrain-AI stations

[Figure]

Figure 3. Temporal comparisons of near-surface volumetric water contents between observations at 5 cm depth and model simulations centered at 3.5 cm depth. The model simulations are contrasted with 20 cm depth due to unavailable near-surface observations for Johnstown Castle and Dripsey

[Figure]

Figure 4. Temporal comparisons of sub-surface volumetric water contents between observations top 20 cm depth and model simulations centered at 17.5 cm depth.

Comment #3:

The satellite observation. Why you chose the ASCAT database as a reference? In my practice, the ESA-CCI or SMAP datasets are usually perform better than ASCAT. Is it because the ASCAT has best performance (use the station observation as a reference) or just ASCAT dataset can reproduce a dry bias pattern? In addition, how do you consider the influence of uncertainties of ASCAT dataset on the evaluation results?

Response #3:

While it may be true that ESA-CCI are a better performing product than ASCAT, this also depends on the types of products employed, for example, ASCAT has achieved the best performance among non-blended products, including ESA-CCI combined products (e.g. Mazzariello et al., 2023).

We have independently evaluated ESACCI SSM (25 km resolution), GSSM (1 km machine learning based surface soil moisture products) (Han et al., 2023) and ASCAT 1 km SWI (linearly transformed using

variance matching with station observations), and carefully chosen the latter as the reference. Evidence from our evaluation of these products against the station observations (Figures 5 and 6 below) suggests that ASCAT yields better performance than ESA CCI SSM and GSSM 1 km products, though the latter products show higher temporal dynamics as shown by the higher temporal correlations with the ground observations.  The rising and falling trends are also better captured by ASCAT.  While the uncertainty in GSSM products is likely linked to lack of training data from Ireland, the biases in ESA CCI SSM may be attributed to its native grid resolution which is too coarse to effectively represent the soil heterogeneity, and/or differences in soil depths.

In addition, the model standard errors of 0-0.07 $m^3$ $m^{-3}$ (Figure 1 above), accounting for model uncertainty, are below the ASCAT uncertainty threshold of 0.1 $m^3$ $m^{-3}$. Therefore, model errors may not be overestimated because of large ASCAT uncertainty. However, because ASCAT products do not account for soil textural properties, the use of a characteristic time length (e.g. T2) without soil texture differentiation may influence our results, as the optimal characteristic time lengths differ for different soil texture categories (de Lange et al., 2008).

Reviewer #2 also agrees that other products can be evaluated to support the analysis. We will revise the paper accordingly to reflect this information.

References

Han, Q., Zeng, Y., Zhang, L. et al. Global long term daily 1 km surface soil moisture dataset with physics informed machine learning. *Sci Data* 10, 101 (2023). https://doi.org/10.1038/s41597-023-02011-7

Mazzariello, A, Albano, R., Lacava, T., et al. Intercomparison of recent microwave satellite soil moisture products on European ecoregions, *J. Hydrology*, 626, 130311 (2023). https://doi.org/10.1016/j.jhydrol.2023.130311

R. de Lange, R. Beck, N. van de Giesen, J. Friesen, A. de Wit and W. Wagner, "Scatterometer-Derived Soil Moisture Calibrated for Soil Texture with a One-Dimensional Water-Flow Model," in *IEEE Transactions on Geoscience and Remote Sensing*, 46, 12, 4041-4049, 2008. doi: 10.1109/TGRS.2008.2000796

[Figure]

Figure 5. Evaluation of satellite-derived 1 km ASCAT-T2 (0-10 cm), 1 km GSSM (0-5 cm) and 25 km ESACCI near-surface soil moisture against the station observations. No available ESACCI SSM grid values for Valentia, and due to ASCAT later year of operation in 2015, no available ASCAT values also for Dripsey.

[Figure]

Figure 6. Evaluation of satellite-derived 1 km ASCAT-T10 (10-30 cm) sub-surface soil moisture against the station observations (20 cm). No sub-surface values for ESACCI and GSSM products.

Comment #4:

The soil moisture or soil moisture anomaly. Although SOILGRID seems to show larger negative bias in modeling soil moisture, it improves the correlation coefficient. An important issue is whether the soil moisture absolute value is more important than the soil moisture anomaly (dynamics)? Actually, the observed soil moisture and model simulated soil moisture are physically different. Model simulation is a mean state of a grid box with specific thickness, while observations only represent a point at a fixed depth.

Response #4:

The soil moisture anomaly may be more important if the interest is in the relative state of soil moisture, rather than the absolute soil moisture values and how they may vary seasonally.

There is clear seasonality to soil moisture in Ireland, thus we have looked at evaluating both absolute soil moisture values at point scale and the relative state at grid scale. We demonstrated that the improved correlation coefficient in SOILGRID is consistent regardless, though the improvement is very small relative to STATSGO. The reason for SOILGRID showing higher negative biases and correlation coefficient is due to uncertainty in PTFs-derived soil parameter values and stronger seasonal effect or increased spatial scale, respectively. We have addressed this issue in our response #5 below.

We acknowledge that due to soil heterogeneity across the landscape, the scale discrepancies between point and model grid data or soil depths may have introduced uncertainties in our results. These issues are out of the scope of the current work. However, we have evaluated the model against the comparable and better performing 1 km ASCAT soil moisture products, to reduce the issues of scale mismatch. But again, these satellite products also have inherent uncertainties associated with them as shown above in our response #3. Hence, we have provided a note of caution in P8L18-20 and P9L34-36. Additionally, we have carried out further analysis and demonstrated that 1 km ASCAT SWI is better performing with ground observations (see Figures 5 and 6 above)

We will revise the paper accordingly to properly acknowledge the potential uncertainties due to different limitations. These will include spatial scale mismatches between point-based observations and model 1 km grids; differences in soil depths between observations and model; uncertainties in soil moisture measurements/satellite products.

Comment #5:

Why the SOILGRID improves the simulation of soil moisture dynamics but increases the dry biases? Some in-depth analysis should be provided. In addition, the differences in soil moisture drought may not

simply related to the simulation of soil moisture absolute values because the soil moisture percentiles are used here. I wonder whether the soil hydraulic conductivity or diffusivity is responsible for the difference here. For example, a higher conductivity can cause a faster response of soil moisture to the water deficit.

Response #5:

Thanks for this comment.  We have further investigated this, as a common practice we calculated the soil moisture anomalies for ground observations and model outputs to remove the potential seasonal effect on model evaluation. The calculation was based on a z-score anomaly over a 35-day moving window where the number of soil moisture samples is greater than 6. While Pearson's correlations for anomalies reduce significantly (ranging between 0.45 and 0.78) compared to the absolute values for both model experiments across the stations, the soil moisture dynamics are still sometimes higher in SOILGRID than the STATSGO. Hence, the improved SOILGRID soil moisture dynamics may be linked to the increased resolution (250 m) soil input data which allowed us to better constrain the model soil heterogeneity in terms of the texture and vertical profiles.

In addition, we agree with the reviewer's observations. The SOILGRID dry biases are evident particularly in areas with higher soil hydraulic conductivity, as demonstrated in Figure 7 below.  These areas are mostly associated with increasing grain size (from Clay to Loam or Loam to Sandy Loam), thereby increasing the pore spaces, faster decline in soil moisture memory and rapid drying, relative to the STATSGO. This is in addition to the significant influence of lower soil field capacity as earlier explained in our response #1.

[Figure]

Figure 7. Spatial characteristics of absolute and difference between STATSGO and SOILGRIDS for soil hydraulic conductivity (KSAT)

---

## Author Comment (AC3)

**Authors' response letter** (hess-2023-304)

The following are our responses to comments by the community

Dear Dr Oluwafemi Adeyeri,

We thank you very much for your constructive comments and suggestions. The comments are very useful to improve the quality of our paper. All concerns and suggestions raised are addressed accordingly. Please find below your comments and our responses. Note that the authors' responses are highlighted in red.

1. P3L25-33: These lines are unclear. It is unclear if the authors are evaluating what you already stated as a setback. Are you inheriting these setbacks into your focus?

Response #1: It is true that there are challenges with accurately representing soil physical properties in land surface models, our focus is to investigate how much this uncertainty would impact the simulated land state conditions and whether using a state-of-the-art soil database such as SOILGRID with PTF would add any value.

2. P5L1: Does this improvement assume no heat flux transfer between soil layers? How is this practicable?

Response #2: The use of a zero thermal boundary condition at the bottom deep soil is common in land surface models and is an option in the NOAH-MP model. It assumes that the thermal gradient is small and does not influence temperature much at the shallow soil depth. But we have used a different physics option provided in the model (see Table 3 of the paper) to consider the thermal regime at the subsurface, in our study.

3. P6L29: I am not sure landuse is static, as notable transformations could occur in 10 years.

Response #3: We agree with your point of view, the static here means it does not change over the course of a year. We will revise this to avoid confusion.

4. P5L3: Does the model account for scale-gap effects considering input data resolutions?

Response #4: Yes, the WRF preprocessing system (WPS) was used to harmonise and generate the geographic files and forcing files at 1 km grid space.

5. P6L10: What are the bases for the classification? What is the method used and what is the rationale behind it?

Response #5: Thanks. The goal was to use a high resolution (e.g. 100 m Corine) input land cover product with 44 classes that better represent Ireland's landscape as inputs. The NOAH-MP model provides a

lookup table of vegetation properties that are linked to only 21 land cover categories based on MODIS land cover, reclassifying Corine to match the MODIS was necessary to easily link the 21 vegetation properties. We have used a simple method that merges the common or almost similar land cover types in terms of attributes to reduce the number of categories.

6.  P6L334: How is the stability of the model ascertained?

Response #6: Thanks. This was determined when the difference between present and previous spin-up soil moisture outputs is small or negligible ( ~ $10^{-3}$). By using the input climatology, this was quickly achieved after 3-5 spin up runs.

7.  P7L4: Are there any uncertainties regarding the station data quality and how are thgey assessed?

Response #7: Thanks. Assessing uncertainties in the station data is beyond the scope of this work; however, we have ensured minimum measurement standards in our Terrain-AI stations, including calibration of TDR sensors.

We will revise the paper accordingly to properly acknowledge the potential uncertainties due to different limitations. These will include spatial scale mismatches between point-based observations and model 1 km grids; differences in soil depths between observations and model; uncertainties in soil moisture measurements/satellite products.

8.  P7L14: What does time for the soil to settle mean?

Response #8: In the process of installing the TDR sensors, including digging and covering holes or trenches where the sensors are buried, the soil-to-sensor contact must be good, with no air gaps. Depending on the soil condition, time is often required after installation to achieve this or for the soil to properly settle around the sensor, ideally reaching saturation.

9.  P7L34: Is this a standard product or the authors develop it? If it is a standard product, it needs referencing. If developed by authors, what method was used for this fusion, bearing in mind the transfer of error from fusion approaches?

Response #9: It's a global satellite product, not developed by the authors. Relevant references have been provided and the link to access the product is also provided in the 'code and data availability' section

10. P8L1: Is this different from previously explained?

Response #10: There are different soil moisture products from ASCAT, the statement here explicitly states that we have used the 1 km daily ASCAT SWI which covers the European region in our study.

11. P8L11: what does extracted at model resolution mean and how was this done?

Response #11: What we mean here is that we extracted modeled values for the stations' colocated 1 km by 1 km grid cells. This was done using the 'terra' R package, by converting the stations' geometry into spatial vector points, then using this to extract the corresponding model grid values.

12. P9L27: Seasonal variability can not be assessed simply from the time series presented. In fact, 4f and g are misleading as they did not show seasonal variabilities.

Response #12: Thanks. This will be revised accordingly.

13. P9L35: This assumption does not warrant generalization. The uncertainty associated with comparing the model areal grid to measurement points should be quantified.

Response #13: Quantifying uncertainty associated with scale mismatch between points and grid is beyond the scope of the current work. However, as the standard practice in model evaluation, we have acknowledged this limitation in P8L18-20.

We will further revise the paper accordingly to properly acknowledge the potential uncertainties due to different limitations. These will include spatial scale mismatches between point-based observations and model 1 km grids; differences in soil depths between observations and model; uncertainties in soil moisture measurements/satellite products.

14. P10L1: I suggest the authors combine error metrics instead of individual ones. Since error metrics are sensitive to differences in precision, it is vital to have a combined metric to account for the different statistical properties of an ideal model performance. Collectively assessing these statistical metrics provides a comprehensive understanding of the performance of each model. This approach provides valuable insights for the overall evaluation of the models. The ensemble method of statistical metrics will further reveal the overall efficacy and reliability of the models.

Response #14: Thanks for your suggestion. We are concerned about not introducing confusion, especially in relation to interpreting a weighted metric. Each metric tells something different about our results, understanding these differences is important for the overall interpretation of the results.

15. Figure 5: It is also important to understand this distribution's mean changes. Also, check if these changes are significant.

Response #15: Thanks. This point has been addressed already in the boxplots presented in Figure 5.

16. Figures 10 and 11: No section in the methodology explicitly addressed how these were calculated or generated.

Response #16: We do not agree with this point. P9L15-22 and Table 4 explicitly explain how the maps of drought categories were generated.

17. Address potential limitations of your study, uncertainties in the results, and sources of error. Acknowledge the challenges and potential biases that might affect the interpretation of the findings.

Response #17: Thanks for your suggestion. This concern will be addressed properly in the paper, as already stated in response #13 above.

18. The manuscript's introduction appears overly extensive, encompassing information tangential to the study objectives. A thorough revision to streamline the introduction and enhance focus is recommended.

Response #18: We do not agree with this point. We believe that the current introduction is concise and provides detailed background and existing knowledge (in a brief literature review) about the topic.

---

## Author Response (AR2)

Authors' response letter (hess-2023-304) The following is a point-to-point response to report #2 by reviewer #1

**Dear Reviewer #1,**

We thank you very much for taking out time to read and provide your very useful comments and suggestions. All questions and comments are addressed accordingly. Please find below your comments and our responses. Note that the authors' responses are highlighted in red

Two Questions:

Unit Consistency Between ASCAT and ESA CCI SM Data:

The ASCAT data provides relative humidity values, while the ESA CCI SM product reports absolute soil moisture values. Could you clarify how you rescaled these datasets to ensure unit consistency for comparison with the observational data?

Thank you for your question. The ESA CCI SM product reports absolute soil moisture values, expressed as a ratio of water to soil volume ( $m^3 m^{-3}$ ), which aligns with the observational data, also presented in  $m^3 m^{-3}$ . In contrast, the ASCAT SWI product represents the degree of soil wetness or dryness, expressed as a degree of saturation (0–100%).

To harmonize the ASCAT SWI with the ESA CCI SM and observational data in terms of units, we applied a variance matching approach (see equation below). This method involves linearly transforming the SWI values using the mean ( $\mu$ ) and standard deviation ( $\sigma$ ) of the observational data (VWC). This is a well-established technique for rescaling the SWI values to match observational data or other products expressed in m3 m-3 (e.g., Paulik et al., 2014; Bauer-Marschallinger et al., 2018).

$$SWI_* = \frac{SWI(t) - \mu_{SWI}}{\sigma_{SWI}} \sigma_{VWC} + \mu_{VWC}$$

We have provided this information in the Appendix P47L13 of the revised manuscript.

**References**

Bauer-Marschallinger, B., Paulik, C., Hochstöger, S., Mistelbauer, T.,; Modanesi, S., Ciabatta, L., Massari, C., Brocca, L., Wagner, W. Soil Moisture from Fusion of Scatterometer and SAR: Closing the Scale Gap with Temporal Filtering. Remote Sensing 2018, 10, 1030, https://doi.org/10.3390/rs10071030

Paulik, C., Dorigo, W., Wagner, W., and Kidd, R.: Validation of the ASCAT Soil Water Index using in situ data from the International Soil Moisture Network, *Int. J. Appl. Earth Obs.*, 30, 1–8, https://doi.org/10.1016/j.jag.2014.01.007, 2014

Hydraulic Effects of Soil Organic Matter in PTF:

Did you account for the hydraulic effects of soil organic matter when using the PTF (Pedotransfer Function) with the SOILGRIDS dataset? Soil organic matter is particularly important for high porosity in surface layers, especially in alpine and cold regions. Its inclusion could significantly impact the accuracy of soil moisture simulations.

Thank you for your question. The reviewer raises a valid point. We did not account for the hydraulic effects of soil organic matter (SOM) in the pedotransfer function (PTF) for the following reasons.

1. Although the Saxton and Rawls (2006) PTF can use SOM data, the input SOM content in the NOAH-MP model is hard-coded as 0.0. Therefore, accounting for the effects of SOM would require adjustments to this portion of the model code or the development of a new soil hydraulic parameterization scheme to incorporate available SOM data, as has been done in other similar applications using the NOAH-MP model (e.g., Sun et al., 2021).

2. The necessary SOM data is not readily available in our case. As noted in the manuscript (P7L11-13), only sand and clay soil proportions at four depth layers are currently available as part of the WRF geographical data fields used in the PTF.

We view the incorporation of SOM as a potential avenue for future research, especially considering the differing opinions on its hydraulic effects on soil moisture. While many studies suggest it is significant, others consider it less important (Sun et al., 2021).

**References**

Sun, J., Chen, Y., Yang, K., Lu, H., Zhao, L., Zheng, D. Influence of Organic Matter on Soil Hydrothermal Processes in the Tibetan Plateau: Observation and Parameterization, J. Hydrometeorology, 22, 2659-2674, https://doi.org/10.1175/JHM-D-21-0059.1 2021

**One Comment:**

I agree with the authors' statement: "the need to develop detailed regionally-derived soil texture characteristics and for better representations of soil properties in LSMs." However, it is important to note that the corresponding model parameterizations (e.g., PTF functions) should also be optimized. For instance, the maximum porosity estimated by current PTF functions is typically below 0.4–0.5 m3 m-3, which limits the model's ability to simulate soil moisture values exceeding this threshold, even when the soil texture is accurately represented. Therefore, I suggest revising the sentence in the abstract to emphasize the need for collaborative improvements in both soil texture data and localized PTF parameterizations.

Thanks very much for the comment. This has been incorporated accordingly.